# SUBTRACTIVE MIXTURE MODELS VIA SQUARING: REPRESENTATION AND LEARNING

**Lorenzo Loconte**[1]*  **Aleksanteri M. Sladek**[2]  **Stefan Mengel**[3]

**Martin Trapp**[2]  **Arno Solin**[2]  **Nicolas Gillis**[4]  **Antonio Vergari**[1]

[1] School of Informatics, University of Edinburgh, UK

[2] Department of Computer Science, Aalto University, Finland

[3] University of Artois, CNRS, Centre de Recherche en Informatique de Lens (CRIL), France

[4] Department of Mathematics and Operational Research, Université de Mons, Belgium

## ABSTRACT

Mixture models are traditionally represented and learned by *adding* several distributions as components. Allowing mixtures to *subtract* probability mass or density can drastically reduce the number of components needed to model complex distributions. However, learning such subtractive mixtures while ensuring they still encode a non-negative function is challenging. We investigate how to learn and perform inference on deep subtractive mixtures by *squaring* them. We do this in the framework of probabilistic circuits, which enable us to represent tensorized mixtures and generalize several other subtractive models. We theoretically prove that the class of squared circuits allowing subtractions can be exponentially more expressive than traditional additive mixtures; and, we empirically show this increased expressiveness on a series of real-world distribution estimation tasks.

## 1 INTRODUCTION

Finite mixture models (MMs) are a staple in probabilistic machine learning, as they offer a simple and elegant solution to model complex distributions by blending simpler ones in a linear combination (McLachlan et al., 2019). The classical recipe to design MMs is to compute a *convex combination* over input components. That is, a MM representing a probability distribution $p$ over a set of random variables $\mathbf{X} = \{X_1, X_2, \ldots, X_D\}$ is usually defined as

$$p(\mathbf{X}) = \sum_{i=1}^K w_i p_i(\mathbf{X}), \quad \text{with} \quad w_i \geq 0, \quad \sum_{i=1}^K w_i = 1, \tag{1}$$

where $w_i$ are the mixture parameters and each component $p_i$ is a mass or density function. This is the case for widely-used MMs such as Gaussian mixture models (GMMs) and hidden Markov models (HMMs) but also mixtures of generative models such as normalizing flows (Papamakarios et al., 2021) and deep mixture models such as probabilistic circuits (PCs, Vergari et al., 2019b).

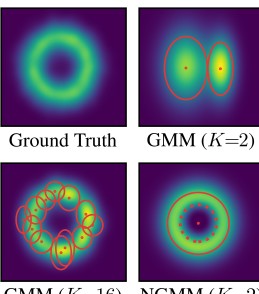

Ground Truth  GMM ($K$=2)

GMM ($K$=16)  NGMM ($K$=2)

The convexity constraint in Eq. (1) is the simplest *sufficient* condition to ensure $p$ is a non-negative function integrating to 1,[1] i.e., is a valid probability distribution, and is often assumed in practice. However, this implies that the components $p_i$ *can only be combined in an additive manner*, and thus it can greatly impact their ability to estimate a distribution efficiently. For instance, consider approximating distributions having "holes" in their domain, such as the simple 2-dimensional ring distribution on the left (ground truth). A classical additive MM such a GMM would ultimately recover it, as it is a universal approximator of density functions (Nguyen et al., 2019), but only by using an unnecessarily high number of components (depicted as red ellipsoids). A MM allowing negative mixture weights, i.e., $w_i < 0$, would instead require

---

*Corresponding author, l.loconte@sms.ed.ac.uk

[1] Across the paper we will abuse the term integration to also refer to summation in case of discrete variables.

only two components, as it can *subtract* one inner Gaussian density from an outer one (NGMM, see dotted ellipsoid). We call these MMs subtractive or *non-monotonic* MMs (NMMs), as opposed to their classical additive counterpart, called *monotonic* MMs (Shpilka & Yehudayoff, 2010).

The challenge with NMMs is ensuring that the modeled $p(\mathbf{X})$ is a valid distribution, as the convexity constraint does not hold anymore. This problem has been investigated in the past in a number of ways, in its simplest form by imposing ad-hoc constraints over the mixture parameters $w_i$, derived for simple components such as Gaussian and Weibull distributions (Zhang & Zhang, 2005; Rabusseau & Denis, 2014; Jiang et al., 1999). However, different families of components would require formulating different constraints, whose closed-form existence is not guaranteed.

In this paper, we study a more general principle to design NMMs that circumvents the aforementioned limitation while ensuring non-negativity of the modeled function: *squaring the encoded linear combination*. For example, the NGMM above is a squared combination of Gaussian densities with negative mixture parameters. We theoretically investigate the expressive efficiency of squared NMMs, i.e., their expressiveness w.r.t. their model size, and show how to effectively represent and learn them in practice. Specifically, we do so in the framework of PCs, tractable models generalizing classical shallow MMs into deep MMs represented as structured neural networks. Deep PCs are already more expressive efficient than shallow MMs as they compactly encode a mixture with an exponential number of components (Jaini et al., 2018; Vergari et al., 2019b). However, they are classically represented with non-negative parameters, hence being restricted to encode deep but additive MMs. Instead, as a main theoretical contribution we prove that *our squared non-monotonic PCs* (NPC$^2$s) *can be exponentially more parameter-efficient than their monotonic counterparts*.

**Contributions.** **i)** We introduce a general framework to represent NMMs via squaring (Sec. 2), within the language of tensorized PCs (Mari et al., 2023), and show how NPC$^2$s can be effectively learned and used for tractable inference (Sec. 3). **ii)** We show how NPC$^2$s generalize not only monotonic PCs but other apparently different models allowing negative parameters that have emerged in different literatures, such as square root of density models in signal processing (Pinheiro & Vidakovic, 1997), positive semi-definite (PSD) models in kernel methods (Rudi & Ciliberto, 2021), and Born machines from quantum mechanics (Orús, 2013; Glasser et al., 2019) (Sec. 4). This allows us to understand why they lead to tractable inference via the property-oriented framework of PCs. **iii)** We derive an exponential lower bound over the size of monotonic PCs to represent functions that can be compactly encoded by one NPC$^2$ (Sec. 4.1), hence showing that NPC$^2$s (and thus the aforementioned models) can be much more expressive for a given size. Finally, **iv)** we provide empirical evidence (Sec. 5) that NPC$^2$s can approximate distributions better than monotonic PCs for a variety of experimental settings involving learning from real-world data and distilling intractable models such as large language models to unlock tractable inference (Zhang et al., 2023).

## 2 SUBTRACTIVE MIXTURES VIA SQUARING

We start by formalizing how to represent *shallow* NMMs by *squaring* non-convex combinations of $K$ simple functions. Like exponentiation in energy-based models (LeCun et al., 2006), squaring ensures the non-negativity of our models, but differently from it, allows to tractably renormalize them. A squared NMM encodes a (possibly unnormalized) distribution $c^2(\mathbf{X})$ over variables $\mathbf{X}$ as

$$c^2(\mathbf{X}) = \left(\sum_{i=1}^K w_i c_i(\mathbf{X})\right)^2 = \sum_{i=1}^K \sum_{j=1}^K w_i w_j c_i(\mathbf{X}) c_j(\mathbf{X}), \tag{2}$$

where $c_i$ are the learnable components and the mixture parameters $w_i \in \mathbb{R}$ are unconstrained, as opposed to Eq. (1). Squared NMMs can therefore represent $\binom{K+1}{2}$ components within the same parameter budget of $K$ components of an additive MM. Each component of a squared NMM computes a *product of experts* $c_i(\mathbf{X}) c_j(\mathbf{X})$ (Hinton, 2002) allowing negative parameters $2w_i w_j$ if $i \neq j$, and $c_i^2(\mathbf{X})$ with $w_i^2$ otherwise. Fig. 1 shows a concrete example of this construction, which constitutes the simplest NPC$^2$ we can build (see Sec. 3), i.e., comprising a single layer and having depth one.

**Tractable marginalization.** Analogously to traditional MMs, squared NMMs support tractable marginalization and conditioning, if their component distributions do as well. The distribution encoded by $c^2(\mathbf{X})$ can be normalized to compute a valid probability distribution $p(\mathbf{X}) = c^2(\mathbf{X})/Z$, by computing its partition function $Z$ as

$$Z = \int c^2(\mathbf{x}) \, \mathrm{d}\mathbf{x} = \sum_{i=1}^K \sum_{j=1}^K w_i w_j \int c_i(\mathbf{x}) c_j(\mathbf{x}) \, \mathrm{d}\mathbf{x}. \tag{3}$$

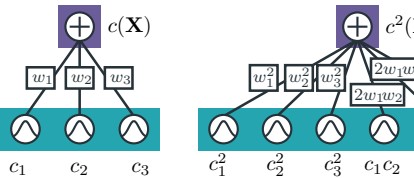

Figure 1: **Shallow MMs and squared NMMs represented as PCs**, mapped to a computational graph having input components and a weighted sum unit as output. Squaring a mixture with $K = 3$ components (left) can yield more components that share parameters (right).

Computing $Z$ translates to evaluating $\binom{K+1}{2}$ integrals over products of components $c_i(\mathbf{X})c_j(\mathbf{X})$. More generally, marginalizing any subset of variables in $\mathbf{X}$ can be done in $\mathcal{O}(K^2)$. This however implies that the components $c_i$ are chosen from a family of functions such that their product $c_i(\mathbf{X})c_j(\mathbf{X})$ can be tractably integrated, and $Z$ is non-zero and finite. This is true for many parametric families, including exponential families (Seeger, 2005). For instance, the product of two Gaussian or two categorical distributions is another Gaussian (Rasmussen & Williams, 2005) or categorical up to a multiplicative factor, which can be computed in polynomial time.

**A wider choice of components.** Note that we do not require each $c_i$ to model a probability distribution, e.g., we might have $c_i(\mathbf{x}) < 0$. This allows us to employ more expressive tractable functions as base components in squared NMMs such as splines (see App. E for details) or potentially small neural networks (see discussion in App. G). However, if the components are already flexible enough there might not be an increase in expressiveness when mixing them in a linear combination or squaring them. E.g., a simple categorical distribution can already capture any discrete distribution with finite support and a (subtractive) mixture thereof might not yield additional benefits besides being easier to learn. An additive mixture of Binomials is instead more expressive than a single Binomial, but expected to be less expressive than its subtractive version (as illustrated in Sec. 5).

**Learning squared NMMs.** The canonical way to learn traditional MMs (Eq. (1)) is by maximum-likelihood estimation (MLE), i.e., by maximizing $\sum_{\mathbf{x} \in \mathcal{D}} \log p(\mathbf{x})$ where $\mathcal{D}$ is a set of independent and identically distributed (i.i.d.) samples. For squared NMMs, the MLE objective is

$$\sum_{\mathbf{x} \in \mathcal{D}} \log \left( c^2(\mathbf{x})/Z \right) = -|\mathcal{D}| \log Z + 2 \sum_{\mathbf{x} \in \mathcal{D}} \log |c(\mathbf{x})|, \tag{4}$$

where $c(\mathbf{x}) = \sum_{i=1}^{K} w_i c_i(\mathbf{x})$. Unlike other NMMs mentioned in Sec. 1, we do not need to derive additional closed-form constraints for the parameters to preserve non-negativity. Although materializing the squared mixture having $\binom{K+1}{2}$ components is required to compute $Z$ as in Eq. (3), evaluating $\log |c(\mathbf{x})|$ is linear in $K$. Hence, we can perform batched stochastic gradient-based optimization and compute $Z$ just once per batch, which makes NMMs efficient to learn (see App. C).

## 3 SQUARING DEEP MIXTURE MODELS

So far, we dealt with mixtures that are shallow, i.e., that can be represented as simple computational graphs with a single weighted sum unit (e.g., Fig. 1). We now generalize them in the framework of PCs (Vergari et al., 2019b; Choi et al., 2020; Darwiche, 2001) as they offer a property-driven language to model structured neural networks which allow tractable inference. PCs enable us to encode an exponential number of mixture components in a compact but deep computational graph.

PCs are usually defined in terms of scalar computational units: sum, product and input (see App. A). Following Vergari et al. (2019a); Mari et al. (2023), we instead formalize them as tensorized computational graphs. That is, we group several computational units together in layers, whose advantage is twofold. First, we are able to derive a simplified tractable algorithm for squaring that requires only linear algebra operations and benefits from GPU acceleration (Alg. 1). Second, we can more easily generalize many recent PC architectures (Peharz et al., 2020b;a; Liu & Van den Broeck, 2021), as well as other tractable tensor representations (Sec. 4). Fig. A.1 illustrates how scalar computational units are mapped to tensorized layers. We start by defining deep computational graphs that can model possibly negative functions, simply named *circuits* (Vergari et al., 2021).

**Definition 1** (Tensorized circuit). A *tensorized circuit* $c$ is a parameterized computational graph encoding a function $c(\mathbf{X})$ and comprising of three kinds of layers: *input*, *product* and *sum*. Each layer comprises computational units defined over the same set of variables, also called its *scope*, and every non-input layer receives input from one or more layers. The scope of each non-input layer is

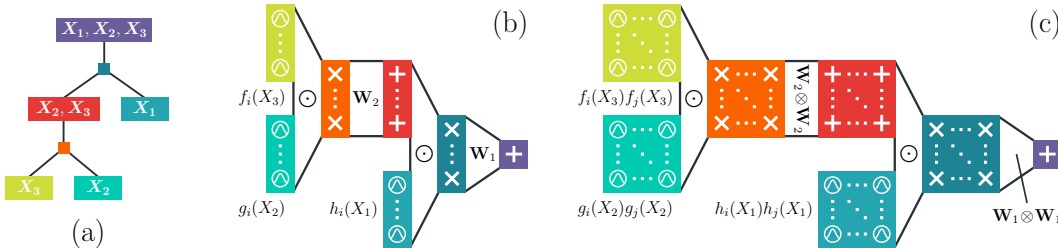

Figure 2: **Squaring tensorized structured-decomposable circuits reduces to squaring layers**, depicted as colored boxes of ⊘ (input), ✕ (product), and ✚ (sum). Connections to a sum layer are labeled by the matrix parameterizing the layer, while connections to product layers are labeled by the Hadamard product sign (see also Fig. A.1). A tensorized structured-decomposable circuit (b) over three variables defined from the RG in (a) is squared in (c) by recursively squaring each layer via Alg. 1. Squared layers contain a quadratic number of units, but still output vectors.

the union of the scope of its inputs, and the scope of the output layer computing $c(\mathbf{X})$ is $\mathbf{X}$. Each input layer $\boldsymbol{\ell}$ has scope $\mathbf{Y} \subseteq \mathbf{X}$ and computes a collection of $K$ functions $f_i(\mathbf{Y}) \in \mathbb{R}$, i.e., $\boldsymbol{\ell}$ outputs a $K$-dimensional vector. Each product layer $\boldsymbol{\ell}$ computes an Hadamard (or element-wise) product over the $N$ layers it receives as input, i.e., $\boldsymbol{\ell} = \odot_{i=1}^{N} \boldsymbol{\ell}_i$. A sum layer with $S$ sum units and receiving input from a previous layer $\boldsymbol{\ell} \in \mathbb{R}^K$, is parameterized by $\mathbf{W} \in \mathbb{R}^{S \times K}$ and computes $\mathbf{W}\boldsymbol{\ell}$.

Fig. 2b shows a deep circuit in tensorized form. To model a distribution via circuits we first require that the output of the computational graph is non-negative. We call such a circuit a PC. Similarly to shallow additive MM (Eq. (1)), a sufficient condition to ensure non-negativity of the output is make the PC monotonic, i.e., to parameterize all sum layers with non-negative matrices and to restrict input layers to encode non-negative functions (e.g., probability mass or density functions). So far, monotonic PCs have been the canonical way to represent and learn PCs (App. G). In Def. 1 we presented product layers computing Hadamard products only, to simplify notation and as this implementation choice is commonly used in many existing PC architectures (Darwiche, 2009; Liu & Van den Broeck, 2021; Mari et al., 2023). We generalize our treatment of PCs in Def. A.6 to deal with another popular product layer implementation: Kronecker products (Peharz et al., 2020b;a; Mari et al., 2023). Our results still hold for both kinds of product layers, if not specified otherwise.

### 3.1 BUILDING TRACTABLE CIRCUITS FOR MARGINALIZATION

Deep PCs can be renormalized and marginalize out any subset of $\mathbf{X}$ in a single feed-forward pass if they are *smooth* and *decomposable*, i.e., each sum layer receives inputs from layers whose units are defined over the same scopes, and each product layer receives inputs from layers whose scopes are pairwise disjoint, respectively (Darwiche, 2001; Choi et al., 2020). See Prop. A.1 for more background. Sum layers in our Def. 1 guarantee smoothness by design as they have exactly one input. A simple way to ensure decomposability is to create a circuit that follows a *hierarchical scope partitioning of variables* $\mathbf{X}$, also called a *region graph*, which is formalized next.

**Definition 2** (Region graph (Dennis & Ventura, 2012))**.** Given a set of variables $\mathbf{X}$, a *region graph* (RG) is a bipartite and rooted graph whose nodes are either *regions*, denoting subsets $\mathcal{R}$ of $\mathbf{X}$, or *partitions* specifying how a region is partitioned into other regions.

Fig. 2a shows an example of a RG. Given a RG, we can build a smooth and decomposable tensorized circuit as follows. First, we parameterize regions $\mathcal{R} \subseteq \mathbf{X}$ that are not further partitioned with an input layer encoding some functions over variables in $\mathcal{R}$. Then, we parameterize each partitioning $\{\mathcal{R}_i\}_{i=1}^{N}$ with a product layer having as inputs one layer for each $\mathcal{R}_i$. Each product layer is then followed by a sum layer. Figs. 2a and 2b illustrate such a construction by color-coding regions and corresponding layers. As we will show in Sec. 3.2, this also provides us a clean recipe to efficiently square a deep circuit. The literature on PCs provides several ways to build RGs (Peharz et al., 2020b;a; Mari et al., 2023). In our experiments (Sec. 5), we recursively partition sets of variables randomly until no further partitioning is possible (Peharz et al., 2020b). Moreover, we experiment with RGs that partitions variables one by one (e.g., the one in Fig. 2a), as they are related to other classes of models (see Sec. 4). App. F further details how to construct these RGs.

## 3.2 SQUARING DEEP TENSORIZED CIRCUITS

**(Squared negative) MMs as circuits.** It is easy to see that traditional shallow MMs (Eq. (1)) can be readily represented as tensorized smooth and decomposable PCs consisting of an input layer encoding $K$ components $p_i$ followed by a sum layer, which is parameterized by a non-negative row-vector $\mathbf{W} \in \mathbb{R}_+^{1 \times K}$ whose entries sum up to one. Squared NMMs (Eq. (2)) can be represented in a similar way, as they can be viewed as mixtures over an increased number of components (see Fig. 1 and Fig. A.1), where the sum layer is parameterized by a vector with real entries, instead. Next, we discuss how to square *deep* tensorized circuits as to retrieve our NPC²s model class.

**Squaring (and renormalizing) tensorized circuits.** The challenge of squaring a tensorized non-monotonic circuit $c$ (potentially encoding a negative function) is guaranteeing $c^2$ to be representable as a smooth and decomposable PC with polynomial size, as these two properties are necessary conditions to being able to renormalize $c^2$ efficiently and in a single feed-forward pass (Choi et al., 2020). In general, even squaring a decomposable circuit while preserving decomposability of the squared circuit is a #P-hard problem (Shen et al., 2016; Vergari et al., 2021). Fortunately, it is possible to obtain a decomposable representation of $c^2$ efficiently for circuits $c$ that are *structured-decomposable* (Pipatsrisawat & Darwiche, 2008; Vergari et al., 2021). Intuitively, in a tensorized structured-decomposable circuit all product layers having the same scope $\mathbf{Y} \subseteq \mathbf{X}$ decompose $\mathbf{Y}$ over their input layers in the exact same way. We formalize this property in the Appendix in Def. A.3.

Tensorized circuits satisfying this property by design can be easily constructed by stacking layers conforming to a RG, as discussed before, and requiring that such a RG is a *tree*, i.e., in which there is a single way to partition each region, and whose input regions do not have overlapping scopes. E.g., the RG in Fig. 2a is a tree RG. From here on, w.l.o.g. we assume our tree RGs to be binary trees, i.e., they partition each region into two other regions only. Given a tensorized structured-decomposable circuit $c$ defined on such a tree RG, Alg. 1 efficiently constructs a smooth and decomposable tensorized circuit $c^2$. Moreover, let $L$ be the number of layers and $M$ the maximum time required to evaluate one layer in $c$, then the following proposition holds.

**Proposition 1** (Tractable marginalization of squared circuits). *Let $c$ be a tensorized structured-decomposable circuit where the products of functions computed by each input layer can be tractably integrated. Any marginalization of $c^2$ obtained via Alg. 1 requires time and space $\mathcal{O}(L \cdot M^2)$.*

See App. B.2 for a proof. In a nutshell, this is possible because Alg. 1 recursively squares each layer $\boldsymbol{\ell}$ in $c$ such as $\boldsymbol{\ell}^2 = \boldsymbol{\ell} \otimes \boldsymbol{\ell}$ in $c^2$, where $\otimes$ denotes the Kronecker product of two vectors.[2] Our tensorized treatment of circuits allows for a much more compact version of the more general algorithm proposed in Vergari et al. (2021) which was defined in terms of squaring scalar computational units. At the same time, it lets us derive a tighter worst-case upper-bound than the one usually reported for squaring structured-decomposable circuits (Pipatsrisawat & Darwiche, 2008; Choi et al., 2015; Vergari et al., 2021), which is the squared number of computations in the whole computational graph, or $\mathcal{O}(L^2 \cdot M^2)$. Note that materializing $c^2$ is needed when we want to efficiently compute the normalization constant $Z$ of $c^2$ or marginalizing any subset of variables. As such, when learning by MLE (Eq. (4)) and by batched gradient descent, we need to evaluate $c^2$ only once per batch, thus greatly amortizing its cost. In App. C, we investigate the time and memory costs of learning NPC²s having different size and on different data set dimensionalities. Finally, tractable marginalization enables tractable sampling from the distribution modeled by NPC²s, as we discuss in App. A.2.

## 3.3 NUMERICALLY STABLE INFERENCE AND LEARNING

Renormalizing deep PCs can easily lead to underflows and/or overflows. In monotonic PCs, this is usually addressed by performing computations in log-space and utilizing the log-sum-exp trick (Blanchard et al., 2021). However, this is not applicable to non-monotonic PCs as intermediate layers can compute negative values. Therefore, we instead evaluate circuits by propagating the logarithm of absolute values and the sign values of the outputs of each layer. Then, sum layers are evaluated with a *sign-aware* version of the log-sum-exp trick. A similar idea has been already applied to evaluate expectations of negative functions with monotonic PCs (Mauá et al., 2018; Correia & de Campos, 2019). App. D extends it to tensorized non-monotonic circuits.

---

[2]In Alg. B.1 we provide a generalization of Alg. 1 to square Kronecker product layers (Peharz et al., 2020b).

---

**Algorithm 1** squareTensorizedCircuit($\ell, \mathcal{R}$)

---

**Input:** A tensorized circuit having output layer $\ell$ and defined on a tree RG rooted by $\mathcal{R}$.
**Output:** The tensorized squared circuit defined on the same tree RG having $\ell^2$ as output layer computing $\ell \otimes \ell$.

1: **if** $\ell$ is an input layer **then**
2:     $\ell$ computes $K$ functions $f_i(\mathcal{R})$
3:     **return** An input layer $\ell^2$ computing all $K^2$
4:         product combinations $f_i(\mathcal{R}) f_j(\mathcal{R})$
5: **else if** $\ell$ is a product layer **then**
6:     $\{(\ell_\text{i}, \mathcal{R}_\text{i}), (\ell_\text{ii}, \mathcal{R}_\text{ii})\} \leftarrow$ getInputs($\ell, \mathcal{R}$)
7:     $\ell_\text{i}^2 \leftarrow$ squareTensorizedCircuit($\ell_\text{i}, \mathcal{R}_\text{i}$)
8:     $\ell_\text{ii}^2 \leftarrow$ squareTensorizedCircuit($\ell_\text{ii}, \mathcal{R}_\text{ii}$)

9:     **return** $\ell_\text{i}^2 \odot \ell_\text{ii}^2$
10: **else**               $\triangleright \ell$ is a sum layer
11:     $\{(\ell_\text{i}, \mathcal{R})\} \leftarrow$ getInputs($\ell, \mathcal{R}$)
12:     $\ell_\text{i}^2 \leftarrow$ squareTensorizedCircuit($\ell_\text{i}, \mathcal{R}$)
13:     $\mathbf{W} \in \mathbb{R}^{S \times K} \leftarrow$ getParameters($\ell$)
14:     $\mathbf{W}' \in \mathbb{R}^{S^2 \times K^2} \leftarrow \mathbf{W} \otimes \mathbf{W}$
15:     **return** $\mathbf{W}' \ell_\text{i}^2$

---

# 4 EXPRESSIVENESS OF NPC²S AND RELATIONSHIP TO OTHER MODELS

Circuits have been used as the "lingua franca" to represent apparently different tractable model representations (Darwiche & Marquis, 2002; Shpilka & Yehudayoff, 2010), and to investigate their ability to exactly represent certain function families with only a polynomial increase in model size – also called the *expressive efficiency* (Martens & Medabalimi, 2014), or *succinctness* (de Colnet & Mengel, 2021) of a model class. This is because the size of circuits directly translates to the computational complexity of performing inference. As we extend the language of monotonic PCs to include negative parameters, here we provide polytime reductions from tractable probabilistic model classes emerging from different application fields that can encode subtractions, to (deep) non-monotonic PCs. By doing so, we not only shed light on why they are tractable, by explicitly stating their structural properties as circuits, but also on why they can be more expressive than classical additive MMs, as we prove that NPC²s can be exponentially more compact in Sec. 4.1.

**Simple shallow NMMs** have been investigated for a limited set of component families, as discussed in Sec. 1. Notably, this can also be done by directly learning to approximate the square root of a density function, as done in signal processing with wavelet functions as components (Daubechies, 1992; Pinheiro & Vidakovic, 1997) or RBF kernels, i.e., unnormalized Gaussians centered over data points (Schölkopf & Smola, 2001), as in Hong & Gao (2021). As discussed in Sec. 3, we can readily represent these NMMs as simple NPC²s where kernel functions are computed by input layers.

**Positive semi-definite (PSD) models** (Rudi & Ciliberto, 2021; Marteau-Ferey et al., 2020) are a recent class of models from the kernel and optimization literature. Given a kernel function $\kappa$ (e.g., an RBF kernel as in Rudi & Ciliberto (2021)) and a set of $d$ data points $\mathbf{x}^{(1)}, \ldots, \mathbf{x}^{(d)}$ with $\boldsymbol{\kappa}(\mathbf{x}) = [\kappa(\mathbf{x}, \mathbf{x}^{(1)}), \ldots, \kappa(\mathbf{x}, \mathbf{x}^{(d)})]^\top \in \mathbb{R}^d$, and a real $d \times d$ PSD matrix $\mathbf{A}$, they define an unnormalized distribution as the non-negative function $f(\mathbf{x}; \mathbf{A}, \boldsymbol{\kappa}) = \boldsymbol{\kappa}(\mathbf{x})^\top \mathbf{A} \boldsymbol{\kappa}(\mathbf{x})$. Although apparently different, they can be translated to NPC²s in polynomial time.

**Proposition 2** (Reduction from PSD models). *A PSD model with kernel function $\kappa$, defined over $d$ data points, and parameterized by a PSD matrix $\mathbf{A}$, can be represented as a mixture of squared NMMs (hence NPC²s) in time $\mathcal{O}(d^3)$.*

We prove this in App. B.3. Note that while PSD models are *shallow* non-monotonic PCs, we can stack them into deeper NPC²s that support tractable marginalization via structured-decomposability.

**Tensor networks and the Born rule.** Squaring a possibly negative function to retrieve an unnormalized distribution is related to the Born rule in quantum mechanics (Dirac, 1930), used to characterize the distribution of particles by squaring their wave function (Schollwoeck, 2010; Orús, 2013). These functions can be represented as a large $D$-dimensional tensor $\mathcal{T}$ over discrete variables $\mathbf{X} = \{X_1, \ldots, X_D\}$ taking value $\{1, \ldots, m\}$, compactly factorized in a tensor network (TN) such as a matrix-product state (MPS) (Pérez-García et al., 2007), also called tensor-train. Given an assignment $\mathbf{x} = \langle x_1, \ldots, x_D \rangle$ to $\mathbf{X}$, a rank $r$ MPS compactly represents $\mathcal{T}$ as

$$\mathcal{T}[x_1, \ldots, x_D] = \sum_{i_1=1}^r \sum_{i_2=1}^r \cdots \sum_{i_{D-1}=1}^r \mathbf{A}_1[x_1, i_1] \mathbf{A}_2[x_2, i_1, i_2] \cdots \mathbf{A}_D[x_D, i_{D-1}], \quad (5)$$

where $\mathbf{A}_1, \mathbf{A}_D \in \mathbb{R}^{m \times r}$, $\mathbf{A}_j \in \mathbb{R}^{m \times r \times r}$ with $1 < j < D$, for indices $\{i_1, \ldots, i_{D-1}\}$, and denoting indexing with square brackets. To encode a distribution $p(\mathbf{X})$, one can reparameterize tensors $\mathbf{A}_j$ to be non-negative (Glasser et al., 2019) or apply the Born rule and square $\mathcal{T}$ to model

$p(\mathbf{x}) \propto (\mathcal{T}[x_1, \ldots, x_D])^2$. Such a TN is called a Born machine (BM) (Glasser et al., 2019). Besides modeling complex quantum states, TNs such as BMs have also been explored as classical ML models to learn discrete distributions (Stoudenmire & Schwab, 2016; Han et al., 2018; Glasser et al., 2019; Cheng et al., 2019), in quantum ML (Liu & Wang, 2018; Huggins et al., 2018), and more recently extended to continuous domains by introducing sets of basis functions, called TTDE (Novikov et al., 2021). Next, we show they are a special case of NPC$^2$s.

**Proposition 3** (Reduction from BMs). *A BM encoding $D$-dimensional tensor with $m$ states by squaring a rank $r$ MPS can be exactly represented as a structured-decomposable* NPC$^2$ *in $\mathcal{O}(D \cdot k^4)$ time and space, with $k \leq \min\{r^2, mr\}$.*

We prove this in App. B.4 by showing an equivalent NPC$^2$ defined on linear tree RG (e.g., the one in Fig. 2a). This connection highlights how tractable marginalization in BMs is possible thanks to structured-decomposability (Proposition 1), a condition that to the best of our knowledge was not previously studied for TNs. Futhermore, as NPC$^2$s we can now design more flexible tree RGs, e.g., randomized tree structures (Peharz et al., 2020b; Di Mauro et al., 2017; Di Mauro et al., 2021), densely tensorized structures heavily exploiting GPU parallelization (Peharz et al., 2020a; Mari et al., 2023) or heuristically learn them from data (Liu & Van den Broeck, 2021).

## 4.1 Exponential Separation of NPC$^2$s and Structured Monotonic PCs

Squaring via Alg. 1 can already make a tensorized (monotonic) PC more expressive, but only by a *polynomial* factor, as we quadratically increase the size of each layer, while keeping the same number of learnable parameters (similarly to the increased number of components of squared NMMs (Sec. 2)). On the other hand, allowing negative parameters can provide an *exponential* advantage, as proven for certain circuits (Valiant, 1979), but understanding if this advantage carries over to our squared circuits is not immediate. In fact, we observe there cannot be any expressiveness advantage in squaring certain classes of non-monotonic structured-decomposable circuits. These are the circuits that support tractable maximum-a-posteriori inference (Choi et al., 2020) and satisfy an additional property known as *determinism* (see Darwiche (2001), Def. A.5). Squaring these circuits outputs a PC of the same size and that is monotonic, as formalized next and proven in App. B.6.

**Proposition 4** (Squaring deterministic circuits). *Let $c$ be a smooth, decomposable and deterministic circuit, possibly computing a negative function. Then, the squared circuit $c^2$ is monotonic and has the same structure (and hence size) of $c$.*

The NPC$^2$s we considered so far, as constructed in Sec. 3, are *not* deterministic. Here we prove that some non-negative functions (hence probability distributions up to renormalization) can be computed by NPC$^2$s that are *exponentially smaller* than any structured-decomposable monotonic PC.

**Theorem 1** (Expressive efficiency of NPC$^2$s). *There is a class of non-negative functions $\mathcal{F}$ over variables $\mathbf{X}$ that can be compactly represented as a shallow squared NMM (hence NPC$^2$s), but for which the smallest structured-decomposable monotonic PC computing any $F \in \mathcal{F}$ has size $2^{\Omega(|\mathbf{X}|)}$.*

We prove this in App. B.5 by showing a non-trivial lower bound on the size of structured-decomposable monotonic PCs for a variant of the unique disjointness problem (Fiorini et al., 2015). Intuitively, this tells us that, given a fixed number of parameters, NPC$^2$s can potentially be much more expressive than structured-decomposable monotonic PCs (and hence shallow additive MMs). We conjecture that an analogous lower bound can be devised for decomposable monotonic PCs. Furthermore, as this result directly extends to PSD and BM models (Sec. 4), it opens up interesting theoretical connections in the research fields of kernel-based and tensor network models.

## 5 Experiments

We aim to answer the following questions: **(A)** are NPC$^2$s better distribution estimators than monotonic PCs? **(B)** how the increased model size given by squaring and the presence of negative parameters independently influence the expressiveness of NPC$^2$s? **(C)** how does the choice of input layers and the RG affect the performance of NPC$^2$s? We perform several distribution estimation experiments on both synthetic and real-world data, and label the following paragraphs with letters denoting relevance to the above questions. Moreover, note that our comparisons between NPC$^2$s and monotonic PCs are based on models having the same number of learnable parameters.

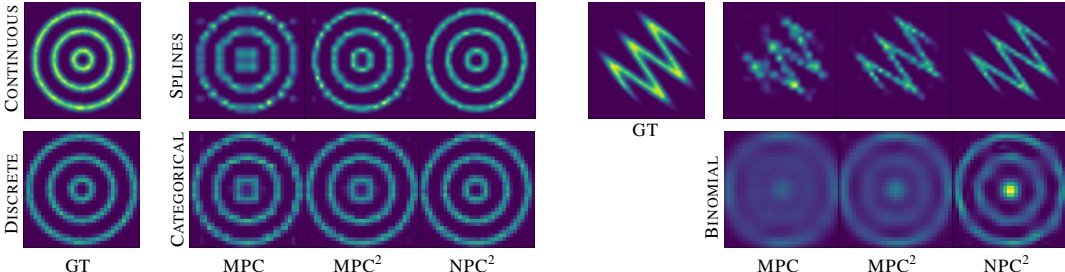

Figure 3: **NPC²s are better estimators, especially with parameter-efficient input layers.** Distribution estimated by monotonic PCs (MPC), squared monotonic PCs (MPC²) and NPC²s on 2D continuous (above) and discrete (below) data. On continuous data input layers compute splines (Eq. (11)), while on discrete data they compute either categoricals (for MPC and MPC²), embeddings (for NPC²s) or Binomials. Apps. H.1 and H.2 shows log-likelihoods on also additional data.

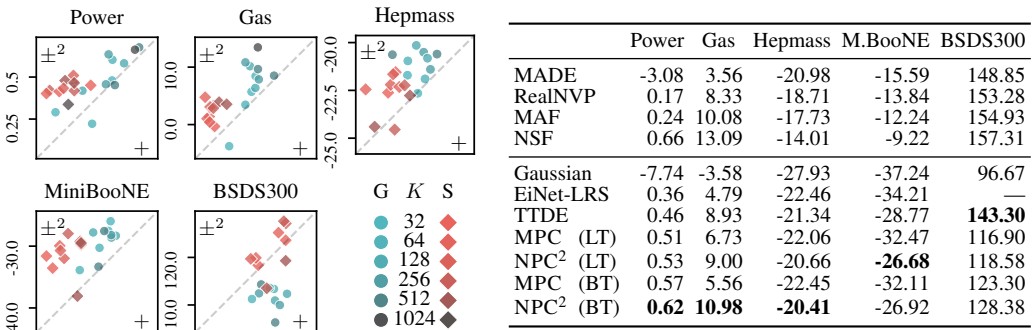

| | Power | Gas | Hepmass | M.BooNE | BSDS300 |
|---|---|---|---|---|---|
| MADE | -3.08 | 3.56 | -20.98 | -15.59 | 148.85 |
| RealNVP | 0.17 | 8.33 | -18.71 | -13.84 | 153.28 |
| MAF | 0.24 | 10.08 | -17.73 | -12.24 | 154.93 |
| NSF | 0.66 | 13.09 | -14.01 | -9.22 | 157.31 |
| Gaussian | -7.74 | -3.58 | -27.93 | -37.24 | 96.67 |
| EiNet-LRS | 0.36 | 4.79 | -22.46 | -34.21 | — |
| TTDE | 0.46 | 8.93 | -21.34 | -28.77 | **143.30** |
| MPC (LT) | 0.51 | 6.73 | -22.06 | -32.47 | 116.90 |
| NPC² (LT) | 0.53 | 9.00 | -20.66 | **-26.68** | 118.58 |
| MPC (BT) | 0.57 | 5.56 | -22.45 | -32.11 | 123.30 |
| NPC² (BT) | **0.62** | **10.98** | **-20.41** | -26.92 | 128.38 |

Legend: G  K  S — ● 32 ◆ ● 64 ◆ ● 128 ◆ ● 256 ◆ ● 512 ◆ ● 1024 ◆

Figure 4: **NPC²s can be more expressive than monotonic PCs (MPCs).** Best average log-likelihoods achieved by monotonic PCs (+) and NPC²s ($\pm^2$), built either from randomized linear tree (LT) or binary tree (BT) RGs (see App. H.3). The scatter plots (left) pairs log-likelihoods based on the number of units per layer $K$ (the higher the darker), differentiating PCs with Gaussian (G/blue) and splines (S/red) input layers. Both axes of each scatter plot are on the same scale, thus the results above the diagonal are of NPC²s achieving higher log-likelihoods than MPCs at parity of model size. The table (right) shows our models' best average test log-likelihoods and puts them in context with intractable (above) and tractable (below) models w.r.t. variable marginalization.

**(A, B) Synthetic continuous data.** Following Wenliang et al. (2019), we evaluate monotonic PCs and NPC²s on 2D density estimation tasks, as this allows us to gain an insight on the learned density functions. To disentangle the effect of squaring versus that of negative parameters, we also experiment with squared monotonic PCs. We build circuit structures from a trivial tree RG (see App. H.1 for details). We experiment with splines as input layers for NPC²s, and enforce their non-negativity for monotonic PCs (see App. E). Fig. 3 shows that, while squaring benefits monotonic PCs, negative parameters in NPC²s are needed to better capture complex target densities.

**(C) Synthetic discrete data.** We estimate the probability mass of the previous 2D data sets, now finitely-discretized (see App. H.2), to better understand when negative parameters might bring little to no advantage if input layers are already expressive enough. First, we experiment with (squared) monotonic PCs (resp. NPC²s) having input layers computing categoricals (resp. real-valued embeddings). Second, we employ the less flexible but more parameter-efficient Binomials instead. App. H.2 reports the hyperparameters. Fig. 3 shows that, while there is no clear advantage for NPC²s equipped with embeddings instead of MPC² with categoricals, in case of Binomials they can better capture the target distribution. This is because categoricals (and embeddings) already have enough parameters to capture "holes" in the probability mass function. However, Binomials introduce a strong inductive bias that might hinder learning. We believe this is the reason why, according to some preliminary results, we did not observe an improvement of NPC²s with respect to monotonic PCs on estimating image distributions.

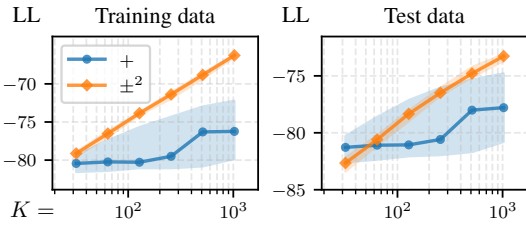

Figure 5: **NPC²s** ($\pm^2$) **achieve higher log-likelihoods** than monotonic PCs ($+$) on data sampled by GPT2. We report the median and the area including 90% of runs by varying the size of layers $K$ and other hyperparameters (App. H.4). For comparison, the log-likelihood of GPT2 on the same training data is about $-52$. The difference on the test data is significant for most values of $K$ (see p-values in Table H.7).

**(A, B, C) Multi-variate continuous data.** Following Papamakarios et al. (2017), we evaluate deeper PCs for density estimation on five multivariate data sets (see Table H.1). We evaluate monotonic PCs and NPC²s in tensorized form built out of randomized linear tree RGs. That is, for some variable permutation, we construct a tree RG where each partition splits a region into a set of only one variable and recursively factorizes the rest. By doing so, we recover an architecture similar to a BM or TTDE (see Sec. 4). Following Peharz et al. (2020b), we also experiment with binary tree RGs whose regions are randomly split in half. App. H.3 details these RGs, as well as the hyperparameters used. We compare against: a full covariance Gaussian, normalizing flows RealNVP (Dinh et al., 2017), MADE (Germain et al., 2015), MAF (Papamakarios et al., 2017) and NSF (Durkan et al., 2019), a monotonic PC with input layers encoding flows (EiNet-LRS) (Sidheekh et al., 2023), and TTDE (Novikov et al., 2021). Fig. 4 shows that NPC²s with Gaussian input layers generally achieve higher log-likelihoods than monotonic PCs on four data sets. Fig. H.3 shows similar results when comparing to squared monotonic PCs, thus providing evidence that negative parameters other than squaring contribute to the expressiveness of NPC²s. Binary tree RGs generally deliver better likelihoods than linear tree ones, especially on Gas, where NPC²s using them outperform TTDE.

**(A) Distilling intractable models.** Monotonic PCs have been used to approximate intractable models such as LLMs and perform exact inference in presence of logical constraints, such as for constrained text generation (Zhang et al., 2023). As generation performance is correlated with how well the LLM is approximated by a tractable model, we are interested in how NPC²s can better be the distillation target of a LLM such as GPT2, rather than monotonic PCs. Following Zhang et al. (2023), we minimize the KL divergence between GPT2 and our PCs on a data set of sampled sentences (details in App. H.4). Since sentences are sequences of token variables, the architecture of tensorized circuits is built from a linear tree RG, thus corresponding to an inhomogeneous HMM in case of monotonic PCs (see App. B.4.1) while resembling a BM for NPC²s. Fig. 5 shows that NPC²s can distill GPT2 and scale better than monotonic PCs, as they achieve log-likelihoods closer to the ones computed by GPT2. We observe that NPC²s fit the training data much better than the test data, even though the results on test data are generally significant (see Table H.7). While this is further evidence of their increased expressiveness, regularizing NPC²s deserves future investigation.

## 6 DISCUSSION & CONCLUSION

With this work, we hope to popularize subtractive MMs via squaring as a simple and effective model class in the toolkit of tractable probabilistic modeling and reasoning that can rival traditional additive MMs. By casting them in the framework of circuits, we presented how to effectively represent and learn deep subtractive MMs such as NPC²s (Sec. 3) while showing how they can generalize other model classes such as PSD and tensor network models (Sec. 4). Our main theoretical result (Sec. 4.1) applies also to these models and justifies the increased performance we found in practice (Sec. 5). This work is the first to rigorously address representing and learning non-monotonic PCs in a general way, and opens up a number of future research directions. The first one is to retrieve a latent variable interpretation for NPC²s, as negative parameters in a non-monotonic PC invalidate the probabilistic interpretation of its sub-circuits (Peharz et al., 2017), making it not possible to learn its structure and parameters in classical ways (see App. G). Better ways to learn NPC²s, in turn, can benefit all applications in which PCs are widely used – from causal discovery (Wang et al., 2022) to variational inference (Shih & Ermon, 2020) and neuro-symbolic AI (Ahmed et al., 2022) – by making more compact and expressive distributions accessible. Finally, by formally connecting circuits with tensor networks, we hope to inspire works that carry over the advancements of one community to the other, such as better learning schemes (Stoudenmire & Schwab, 2016; Novikov et al., 2021), and more flexible ways to factorize high-dimensional tensors (Mari et al., 2023).

REPRODUCIBILITY STATEMENT

In App. H we include all the details about the experiments we showed in Sec. 5. The source code, documentation, data sets and scripts needed to reproduce the results and figures, are available at https://github.com/april-tools/squared-npcs.

ACKNOWLEDGMENTS

AV was supported by the "UNREAL: Unified Reasoning Layer for Trustworthy ML" project (EP/Y023838/1) selected by the ERC and funded by UKRI EPSRC. NG acknowledges the support by the European Union (ERC consolidator, eLinoR, no 101085607). AMS acknowledges funding from the Helsinki Institute for Information Technology. MT acknowledges funding from the Research Council of Finland (grant number 347279). AS acknowledges funding from the Research Council of Finland (grant number 339730). We acknowledge CSC – IT Center for Science, Finland, for awarding this project access to the LUMI supercomputer, owned by the EuroHPC Joint Undertaking, hosted by CSC (Finland) and the LUMI consortium through CSC. We acknowledge the computational resources provided by the Aalto Science-IT project.

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

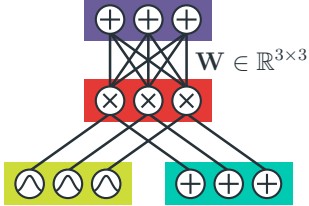 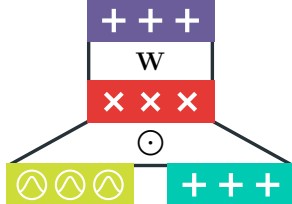

Figure A.1: **Computational units can be grouped into layers as to build a tensorized circuit**. Sum units each parameterized by the rows of $\mathbf{W} \in \mathbb{R}^{3 \times 3}$ (left, in purple) form a sum layer parameterized by $\mathbf{W}$ (right). Product units (left, in red) form an Hadamard product layer (right). Input units (left, in yellow) form an input layer computing the same functions (right).

## A  CIRCUITS

In Sec. 3 we introduced circuits in a tensorized formalism. Here we instead present the definitions and properties of circuits as they are usually defined in the literature, which will be used in App. B.

**Definition A.1** (Circuit (Choi et al., 2020; Vergari et al., 2021)). A *circuit* $c$ is a parameterized computational graph over variables $\mathbf{X}$ encoding a function $c(\mathbf{X})$ and comprising three kinds of computational units: *input*, *product*, and *sum*. Each product or sum unit $n$ receives as inputs the outputs of other units, denoted with the set $\mathsf{in}(n)$. Each unit $n$ encodes a function $c_n$ defined as: (i) $f_n(\mathsf{sc}(n))$ if $n$ is an input unit, where $f_n$ is a function over variables $\mathsf{sc}(n) \subseteq \mathbf{X}$, called its *scope*, (ii) $\prod_{i \in \mathsf{in}(n)} c_i(\mathsf{sc}(n_i))$ if $n$ is a product unit, and (iii) $\sum_{i \in \mathsf{in}(n)} w_i c_i(\mathsf{sc}(n_i))$ if $n$ is a sum unit, with $w_i \in \mathbb{R}$ denoting the weighted sum parameters. The scope of a product or sum unit $n$ is the union of the scopes of its inputs, i.e., $\mathsf{sc}(n) = \bigcup_{i \in \mathsf{in}(n)} \mathsf{sc}(i)$.

Note that tensorized circuits (Def. 1) are circuits where each input (resp. product and sum) layer consists of scalar input (resp. product and sum) units. For example, Fig. A.1 shows how computational units are grouped into layers. A *probabilistic circuit* (PC) is defined as a circuit encoding a non-negative function. PCs that are *smooth* and *decomposable* (Def. A.2) enable computing the partition function and, more in general, performing variable marginalization efficiently (Prop. A.1).

**Definition A.2** (Smoothness and decomposability (Darwiche & Marquis, 2002)). A circuit is *smooth* if for every sum unit $n$, its input units depend all on the same variables, i.e, $\forall i, j \in \mathsf{in}(n)\colon \mathsf{sc}(i) = \mathsf{sc}(j)$. A circuit is *decomposable* if the inputs of every product unit $n$ depend on disjoint sets of variables, i.e, $\forall i, j \in \mathsf{in}(n)\ i \neq j\colon \mathsf{sc}(i) \cap \mathsf{sc}(j) = \varnothing$.

**Proposition A.1** (Tractability (Choi et al., 2020)). Let $c$ be a smooth and decomposable circuit over variables $\mathbf{X}$ whose input units can be integrated efficiently. Then for any $\mathbf{Z} \subseteq \mathbf{X}$ and $\mathbf{y}$ an assignment to variables in $\mathbf{X} \setminus \mathbf{Z}$, the quantity $\int c(\mathbf{y}, \mathbf{z}) \, \mathrm{d}\mathbf{z}$ can be computed exactly in time and space $\Theta(|c|)$, where $|c|$ denotes the size of the circuit, i.e., the number of connections in the computational graph.

The size of circuits in tensorized form is obtained by counting the number of connections between the scalar computational units (see App. A.1.1). Squaring circuits or their tensorized representation efficiently such that the resulting PC is smooth and decomposable (Def. A.2) requires the satisfaction of *structured-decomposability*, as showed in (Pipatsrisawat & Darwiche, 2008; Vergari et al., 2021).

**Definition A.3** (Structured-decomposability (Pipatsrisawat & Darwiche, 2008; Darwiche, 2009)). A circuit is *structured-decomposable* if (1) it is smooth and decomposable, and (2) any pair of product units $n, m$ having the same scope decompose their scope at their input units in the same way.

Note that shallow MMs are both decomposable and structured-decomposable. As anticipated in Sec. 3, the expressiveness of squared non-monotonic PCs that are also *deterministic* is the same as monotonic deterministic PCs, which are used for tractable maximum-a-posteriori (MAP) inference. We prove it formally in App. B.6 by leveraging the definition of determinism that we show in Def. A.5. Before that, we introduce the definition of *support* of a computational unit.

**Definition A.4** (Support (Choi et al., 2020)). The *support* of a computational unit $n$ over variables $\mathbf{X}$ is defined as the set of value assignments to variables in $\mathbf{X}$ such that the output of $n$ is non-zero, i.e., $\mathsf{supp}(n) = \{\mathbf{x} \in \mathsf{val}(\mathbf{X}) \mid c_n(\mathbf{x}) \neq 0\}$, where $\mathsf{val}(\mathbf{X})$ denotes the domain of variables $\mathbf{X}$.

**Definition A.5** (Determinism (Darwiche, 2001)). A circuit $c$ is *deterministic* if for any sum unit $n \in c$ its inputs have disjoint *support* (Def. A.4), i.e., $\forall i, j \in \mathsf{in}(n), i \neq j \colon \mathsf{supp}(i) \cap \mathsf{supp}(j) = \varnothing$.

## A.1 TENSORIZED CIRCUITS

Def. 1 can be further generalized by introducing Kronecker product layers, which are the building blocks of other tensorized circuit architectures, such as randomized and tensorized sum-product networks (RAT-SPNs) (Peharz et al., 2020b), einsum networks (EiNets) (Peharz et al., 2020a).

**Definition A.6** (Tensorized circuit). A *tensorized circuit* $c$ is a parameterized computational graph encoding a function $c(\mathbf{X})$ and comprising of three kinds of layers: *input*, *product* and *sum*. Each layer comprises computational units defined over the same set of variables, also called its *scope*, and every non-input layer receives input from one or more layers. The scope of each non-input layer is the union of the scope of its inputs, and the scope of the output layer computing $c(\mathbf{X})$ is $\mathbf{X}$. Each input layer $\boldsymbol{\ell}$ has scope $\mathbf{Y} \subseteq \mathbf{X}$ and computes a collection of $K$ functions $f_i(\mathbf{Y}) \in \mathbb{R}$, i.e., $\boldsymbol{\ell}$ outputs a $K$-dimensional vector. Each product layer $\boldsymbol{\ell}$ computes either an Hadamard (or element-wise) or Kronecker product over the $N$ layers it receives as input, i.e., $\boldsymbol{\ell} = \odot_{i=1}^N \boldsymbol{\ell}_i$ or $\otimes_{i=1}^N \boldsymbol{\ell}_i$, respectively. A sum layer with $S$ sum units and receiving input form a previous layer $\boldsymbol{\ell} \in \mathbb{R}^K$, is parameterized by $\mathbf{W} \in \mathbb{R}^{S \times K}$ and computes $\mathbf{W}\boldsymbol{\ell}$.

### A.1.1 SIZE OF TENSORIZED CIRCUITS

The time and space complexity of evaluating a circuit is linear in its size. The size $|c|$ of a circuit $c$ (Def. A.1) is obtained by counting the number of input connections of each scalar product or sum unit. In other words, it is the number of edges in the computational graph.

If $c$ is a tensorized circuit, then its size is obtained by counting the number of connections in its non-tensorized form. Fig. A.1 shows part of a tensorized circuit and its non-tensorized form. For sum layers, the number of scalar input connections is the size of its parameterization matrix, i.e., $S \cdot K$ if it is parameterized by $\mathbf{W} \in \mathbb{R}^{S \times K}$. If $\boldsymbol{\ell}$ is an Hadamard product layer computing $\boldsymbol{\ell} = \odot_{i=1}^N \boldsymbol{\ell}_i$, where each $\boldsymbol{\ell}_i$ outputs a $K$-dimensional vector, then the number of its scalar input connections is $N \cdot K$. In case of Kronecker product layers as in the more general Def. A.6, i.e., $\boldsymbol{\ell} = \otimes_{i=1}^N \boldsymbol{\ell}_i$ where each $\boldsymbol{\ell}_i$ outputs a $K$-dimensional vector, then the number of its scalar input connections is $K^{N+1}$.

## A.2 TRACTABLE EXACT SAMPLING

Each sum unit in a monotonic PC can be interpreted as a finitely discrete latent variable that can assume as many values as the number of input connections (Peharz et al., 2017). As such, a monotonic PC can be seen as a hierarchical MM. This allows us to sample exactly from the modeled distribution by (1) recursively sampling latent variables until input units are reached, and (2) sampling observed variables from the distributions modeled by input units (Vergari et al., 2019a).

Such probabilistic interpretation of *inner* sum units for NPC$^2$s is not possible, as they can output negative values. However, since NPC$^2$s are smooth and decomposable (Def. A.2), they support efficient marginalization and hence conditioning (Proposition 1). This allows us to still sample exactly from the modeled distribution via *inverse transform sampling*. That is, we choose a variable ordering $X_1, X_2, \ldots, X_D$ and sample them in an autoregressive fashion, i.e., $x_1 \sim p(X_1)$, $x_2 \sim p(X_2 \mid x_1), \ldots, x_D \sim p(X_D \mid x_1, \ldots, x_{D-1})$, which is still linear in the number of variables.

# B PROOFS

## B.1 SQUARING TENSORIZED CIRCUITS

**Proposition B.1** (Correctness of Alg. 1). Let $c$ be a tensorized structured-decomposable circuit (Def. 1 or its generalization in Def. A.3), then Alg. 1 recursively constructs the layers of the squared tensorized PC $c^2$ such that $c^2$ is also structured-decomposable.

*Proof.* The proof is by induction on the structure of $c$. Let $\boldsymbol{\ell}$ be a sum layer having as input $\boldsymbol{\ell}_i$ and computing $\mathbf{W}\boldsymbol{\ell}_i$, with $\mathbf{W} \in \mathbb{R}^{S \times K}$ and $\boldsymbol{\ell}_i$ computing an output in $\mathbb{R}^S$. If $\boldsymbol{\ell}$ is the last layer of $c$ (i.e.,

the output layer), then $S = 1$ since $c$ outputs a scalar, and the squared layer $\boldsymbol{\ell}^2$ must compute

$$\boldsymbol{\ell}^2 = (\mathbf{W}\boldsymbol{\ell}_{\mathsf{i}}) \cdot (\mathbf{W}\boldsymbol{\ell}_{\mathsf{i}}) = (\mathbf{W} \otimes \mathbf{W})(\boldsymbol{\ell}_{\mathsf{i}} \otimes \boldsymbol{\ell}_{\mathsf{i}}) = (\mathbf{W} \otimes \mathbf{W})\boldsymbol{\ell}_{\mathsf{i}}^2,$$

which requires squaring the input layer $\boldsymbol{\ell}_{\mathsf{i}}$. By inductive hypothesis the squared circuit having $\boldsymbol{\ell}_{\mathsf{i}}^2$ as output layer is structured-decomposable, hence also the squared circuit having $\boldsymbol{\ell}^2$ as output layer must be. If $\boldsymbol{\ell}$ is a non-output sum layer, we still require computing the Kronecker product of its input layer. The squared layer $\boldsymbol{\ell}^2$ is again a sum layer that outputs a $S^2$-dimensional vector, i.e.,

$$\boldsymbol{\ell}^2 = \boldsymbol{\ell} \otimes \boldsymbol{\ell} = (\mathbf{W}\boldsymbol{\ell}_{\mathsf{i}}) \otimes (\mathbf{W}\boldsymbol{\ell}_{\mathsf{i}}) = (\mathbf{W} \otimes \mathbf{W})(\boldsymbol{\ell}_{\mathsf{i}} \otimes \boldsymbol{\ell}_{\mathsf{i}}) = (\mathbf{W} \otimes \mathbf{W})\boldsymbol{\ell}_{\mathsf{i}}^2$$

via mixed-product property (L11-15 in Alg. 1). Let $\boldsymbol{\ell}$ be a binary[3] Hadamard product layer computing $\boldsymbol{\ell}_{\mathsf{i}} \odot \boldsymbol{\ell}_{\mathsf{ii}}$ for input layers $\boldsymbol{\ell}_{\mathsf{i}}, \boldsymbol{\ell}_{\mathsf{ii}}$ each computing a $K$-dimensional vector. Then, the squared layer $\boldsymbol{\ell}^2$ computes the Hadamard product between $K^2$-dimensional vectors, i.e.,

$$\boldsymbol{\ell}^2 = (\boldsymbol{\ell}_{\mathsf{i}} \odot \boldsymbol{\ell}_{\mathsf{ii}}) \otimes (\boldsymbol{\ell}_{\mathsf{i}} \odot \boldsymbol{\ell}_{\mathsf{ii}}) = (\boldsymbol{\ell}_{\mathsf{i}} \otimes \boldsymbol{\ell}_{\mathsf{i}}) \odot (\boldsymbol{\ell}_{\mathsf{ii}} \otimes \boldsymbol{\ell}_{\mathsf{ii}}) = \boldsymbol{\ell}_{\mathsf{i}}^2 \odot \boldsymbol{\ell}_{\mathsf{ii}}^2$$

via mixed-product property with respect to the Hadamard product. By inductive hypothesis $\boldsymbol{\ell}_{\mathsf{i}}^2$ and $\boldsymbol{\ell}_{\mathsf{ii}}^2$ are the output layers of structured-decomposable circuits depending on a disjoint sets of variables. As such, the circuit having $\boldsymbol{\ell}^2$ as output layer maintains structured-decomposability (L6-9 in Alg. 1). For the base case we consider the squaring of an input layer $\boldsymbol{\ell}$ that computes $K$ functions $f_i$ over some variables $\mathbf{Y} \subseteq \mathbf{X}$. We replace $\boldsymbol{\ell}$ with its squaring $\boldsymbol{\ell}^2$ which encodes the products $f_i(\mathbf{Y})f_j(\mathbf{Y})$, $1 \leq i, j \leq K$, by introducing $K^2$ functions $g_{ij}$ such that $g_{ij}(\mathbf{Y}) = f_i(\mathbf{Y})f_j(\mathbf{Y})$ (L2-4 in Alg. 1).

**Squaring Kronecker product layers.** In the case of $\boldsymbol{\ell}$ being a binary Kronecker product layer instead as in the more general Def. A.6, then the squared layer $\boldsymbol{\ell}^2$ computes the Kronecker product between $K^2$-dimensional vectors up to a permutation of the entries, i.e.,

$$\boldsymbol{\ell}^2 = (\boldsymbol{\ell}_{\mathsf{i}} \otimes \boldsymbol{\ell}_{\mathsf{ii}}) \otimes (\boldsymbol{\ell}_{\mathsf{i}} \otimes \boldsymbol{\ell}_{\mathsf{ii}}) = \mathbf{R}\left((\boldsymbol{\ell}_{\mathsf{i}} \otimes \boldsymbol{\ell}_{\mathsf{i}}) \otimes (\boldsymbol{\ell}_{\mathsf{ii}} \otimes \boldsymbol{\ell}_{\mathsf{ii}})\right) = \mathbf{R}\left(\boldsymbol{\ell}_{\mathsf{i}}^2 \otimes \boldsymbol{\ell}_{\mathsf{ii}}^2\right), \tag{6}$$

by introducing a $K^4 \times K^4$ permutation matrix $\mathbf{R}$ whose rows are all zeros except for one entry set to 1, which reorders the entries of $\boldsymbol{\ell}_{\mathsf{i}}^2 \otimes \boldsymbol{\ell}_{\mathsf{ii}}^2$ as to recover the equality in Eq. (6). Note that such permutation maintains decomposability (Def. A.2), and its application can be computed by a sum layer having $\mathbf{R}$ as fixed parameters. Moreover, by inductive hypothesis, the squaring circuit having $\boldsymbol{\ell}^2$ as output layer is still structured-decomposable. Finally, Alg. B.1 generalizes Alg. 1 as to support the squaring of Kronecker product layers as showed above (L10-11 in Alg. B.1). □

---

**Algorithm B.1** squareTensorizedCircuit($\boldsymbol{\ell}, \mathcal{R}$)

**Input:** A tensorized circuit (Def. A.6) having output layer $\boldsymbol{\ell}$ and defined on a tree RG rooted by $\mathcal{R}$.
**Output:** The tensorized squared circuit defined on the same tree RG having $\boldsymbol{\ell}^2$ as output layer computing $\boldsymbol{\ell} \otimes \boldsymbol{\ell}$.

1: **if** $\boldsymbol{\ell}$ is an input layer **then**
2:     $\boldsymbol{\ell}$ computes $K$ functions $f_i(\mathcal{R})$
3:     **return** An input layer $\boldsymbol{\ell}^2$ computing all $K^2$
4:         product combinations $f_i(\mathcal{R})f_j(\mathcal{R})$
5: **else if** $\boldsymbol{\ell}$ is a product layer **then**
6:     $\{(\boldsymbol{\ell}_{\mathsf{i}}, \mathcal{R}_{\mathsf{i}}), (\boldsymbol{\ell}_{\mathsf{ii}}, \mathcal{R}_{\mathsf{ii}})\} \leftarrow \mathsf{getInputs}(\boldsymbol{\ell}, \mathcal{R})$
7:     $\boldsymbol{\ell}_{\mathsf{i}}^2 \leftarrow \mathsf{squareTensorizedCircuit}(\boldsymbol{\ell}_{\mathsf{i}}, \mathcal{R}_{\mathsf{i}})$
8:     $\boldsymbol{\ell}_{\mathsf{ii}}^2 \leftarrow \mathsf{squareTensorizedCircuit}(\boldsymbol{\ell}_{\mathsf{ii}}, \mathcal{R}_{\mathsf{ii}})$
9:     **if** $\boldsymbol{\ell} = \boldsymbol{\ell}_{\mathsf{i}} \odot \boldsymbol{\ell}_{\mathsf{ii}}$ **then return** $\boldsymbol{\ell}_{\mathsf{i}}^2 \odot \boldsymbol{\ell}_{\mathsf{ii}}^2$
10:     **else return** $\mathbf{R}\left(\boldsymbol{\ell}_{\mathsf{i}}^2 \otimes \boldsymbol{\ell}_{\mathsf{ii}}^2\right)$, where $\mathbf{R}$ is
11:       a permutation matrix (see proof of Prop. B.1)
12: **else**         ▷ $\boldsymbol{\ell}$ is a sum layer
13:     $\{(\boldsymbol{\ell}_{\mathsf{i}}, \mathcal{R})\} \leftarrow \mathsf{getInputs}(\boldsymbol{\ell}, \mathcal{R})$
14:     $\boldsymbol{\ell}_{\mathsf{i}}^2 \leftarrow \mathsf{squareTensorizedCircuit}(\boldsymbol{\ell}_{\mathsf{i}}, \mathcal{R})$
15:     $\mathbf{W} \in \mathbb{R}^{S \times K} \leftarrow \mathsf{getParameters}(\boldsymbol{\ell})$
16:     $\mathbf{W}' \in \mathbb{R}^{S^2 \times K^2} \leftarrow \mathbf{W} \otimes \mathbf{W}$
17:     **return** $\mathbf{W}'\boldsymbol{\ell}_{\mathsf{i}}^2$

---

### B.2 Tractable Marginalization of NPC$^2$s

**Proposition 1.** Let $c$ be a tensorized structured-decomposable circuit where the products of functions computed by each input layer can be tractably integrated. Any marginalization of $c^2$ obtained via Alg. 1 requires time and space $\mathcal{O}(L \cdot M^2)$, where $L$ is the number of layers in $c$ and $M$ is the maximum time required to evaluate one layer in $c$ (as detailed in App. A.1.1).

---

[3]Without loss of generality, we assume product layers have exactly two layers as inputs.

*Proof.* Given $c$ by hypothesis, Prop. B.1 ensures that the PC built via Alg. 1 computes $c^2$ and is defined on the same tree RG (Def. 2) of $c$. As such, $c^2$ is structured-decomposable and hence also smooth and decomposable (see Def. A.3). Now, we make an argument about $c$ and $c^2$ in their non-tensorized form (Def. A.1) as to leverage Prop. A.1 for tractable marginalization later. The size of $c$ is $|c| \in \mathcal{O}(L \cdot M)$, where $L$ is the number of layers and $M$ the maximum number of scalar input connections of each layer in $c$ (see App. A.1.1 for details). The size of $c^2$ is therefore $|c^2| \in \mathcal{O}(L \cdot M^2)$, since Alg. 1 squares the output dimension of each layer as well as the size of the parameterization matrix of each sum layer. Since $c^2$ is smooth and decomposable and the functions computed by its input layers can be tractably integrated, then Prop. A.1 ensures we can marginalize any subset of variables in time and space $|c^2| \in \mathcal{O}(L \cdot M^2)$. □

### B.3 REPRESENTING PSD MODELS WITHIN THE LANGUAGE OF NPC$^2$S

**Proposition 2.** A PSD model with kernel function $\kappa$, defined over $d$ data points, and parameterized by a PSD matrix $\mathbf{A}$, can be represented as a mixture of squared NMMs (hence NPC$^2$s) in time $\mathcal{O}(d^3)$.

*Proof.* The PSD model computes a non-negative function $f(\mathbf{x}; \mathbf{A}, \kappa) = \kappa(\mathbf{x})^\top \mathbf{A} \kappa(\mathbf{x})$, where $\kappa(\mathbf{x}) = [\kappa(\mathbf{x}, \mathbf{x}^{(1)}), \dots, \kappa(\mathbf{x}, \mathbf{x}^{(d)})] \in \mathbb{R}^d$, with data points $\mathbf{x}^{(1)}, \dots, \mathbf{x}^{(d)}$, and $\mathbf{A} \in \mathbb{R}^{d \times d}$ is PSD. Let $\mathbf{A} = \sum_{i=1}^r \lambda_i \mathbf{u}_i \mathbf{u}_i^\top$ be the eigendecomposition of $\mathbf{A}$ with rank $r$. Then we can rewrite $f(\mathbf{x}; \mathbf{A}, \kappa)$ as

$$f(\mathbf{x}; \mathbf{A}, \kappa) = \kappa(\mathbf{x})^\top \left( \sum_{i=1}^r \lambda_i \mathbf{u}_i \mathbf{u}_i^\top \right) \kappa(\mathbf{x}) = \sum_{i=1}^r \lambda_i \left( \mathbf{u}_i^\top \kappa(\mathbf{x}) \right)^2,$$

where $\lambda_i > 0$ are positive eigenvalues. Therefore, such PSD model can be represented as a monotonic mixture of $r \le d$ squared NMMs (Eq. (2)), whose $d$ components computing $\kappa(\mathbf{x})$ are shared. The eigendecomposition of $\mathbf{A}$ can be done in time $\mathcal{O}(d^3)$, and materializing each squared NMMs (e.g., as in Fig. 1) requires time and space $\mathcal{O}(d^2)$. Note that if $\mathbf{A} = \mathbf{u}\mathbf{u}^\top$ is a rank-1 matrix, then $f(\mathbf{x}; \mathbf{A}, \kappa) = \left( \mathbf{u}^\top \kappa(\mathbf{x}) \right)^2$ is exactly a squared NMM whose $d$ components compute $\kappa(\mathbf{x})$. □

### B.4 RELATIONSHIP WITH TENSOR NETWORKS

In this section, we detail the construction of a tensorized structured-decomposable circuit (Def. 1) that is equivalent to a matrix product state (MPS) tensor network (Pérez-García et al., 2007), as we mention in Sec. 4. As such, the application of the Born rule as to retrieve a probabilistic model called Born machine (BM) (Glasser et al., 2019) is equivalent to squaring the equivalent circuit (Sec. 3).

**Proposition 3.** A BM encoding $D$-dimensional tensor with $m$ states by squaring a rank $r$ MPS can be exactly represented as a structured-decomposable NPC$^2$ in $\mathcal{O}(D \cdot k^4)$ time and space, with $k \le \min\{r^2, mr\}$.

*Proof.* We prove it constructively, by using a similar transformation used by Glasser et al. (2019) to represent a non-negative MPS factorization as an hidden Markov model (HMM). Let $\mathbf{X} = \{X_1, \dots, X_D\}$ be a set of discrete variables each taking values in $\{1, \dots, m\}$. Let $\mathcal{T}$ be a tensor with $D$ $m$-dimensional indices. Given an assignment $\mathbf{x} = \langle x_1, \dots, x_D \rangle$ to $\mathbf{X}$, we factorize $\mathcal{T}$ via a rank $r$ MPS factorization, i.e.,

$$\mathcal{T}[x_1, \dots, x_D] = \sum_{i_1=1}^r \sum_{i_2=1}^r \cdots \sum_{i_{D-1}=1}^r \mathbf{A}_1[x_1, i_1] \mathbf{A}_2[x_2, i_1, i_2] \cdots \mathbf{A}_D[x_D, i_{D-1}] \quad (7)$$

where $\mathbf{A}_1, \mathbf{A}_D \in \mathbb{R}^{m \times r}$ and $\mathbf{A}_j \in \mathbb{R}^{m \times r \times r}$ with $1 < j < D$, for indices $\{i_1, \dots, i_{D-1}\}$ and denoting indexing with square brackets. To reduce $\mathcal{T}$ to being computed by a tensorized structured-decomposable circuit $c$, i.e., such that $c(\mathbf{x}) = \mathcal{T}[x_1, \dots, x_D]$ for any $\mathbf{x}$, we perform the following construction. First, we perform a *canonical polyadic* (CP) decomposition (Kolda & Bader, 2009) of each $\mathbf{A}_j$ with $1 < j < D$, i.e.,

$$\mathbf{A}_j[x_j, i_{j-1}, i_j] = \sum_{s_j=1}^k \mathbf{B}_j[i_{j-1}, s_j] \mathbf{V}_j[x_j, s_j] \mathbf{C}_j[i_j, s_j]$$

where $k \leq \min\{r^2, mr\}$ is the maximum rank of the CP decomposition (Kolda & Bader, 2009), and $\mathbf{V}_j \in \mathbb{R}^{m \times k}$, $\mathbf{B}_j \in \mathbb{R}^{r \times k}$, $\mathbf{C}_j \in \mathbb{R}^{r \times k}$. Then, we "contract" each $\mathbf{C}_j$ with $\mathbf{B}_{j+1}$ by computing

$$\mathbf{W}_j[s_j, s_{j+1}] = \sum_{i_j=1}^{r} \mathbf{C}_j[i_j, s_j] \mathbf{B}_{j+1}[i_j, s_{j+1}]$$

with $\mathbf{W}_j \in \mathbb{R}^{k \times k}$ for $1 < j < D-1$. In addition, we "contract" $\mathbf{C}_{D-1}$ with $\mathbf{A}_D$ by computing

$$\mathbf{V}_D[x_D, s_{D-1}] = \sum_{i_{D-1}=1}^{r} \mathbf{C}_{D-1}[i_{D-1}, s_{D-1}] \mathbf{A}_D[x_D, i_{D-1}].$$

In addition, for notation clarity we rename $\mathbf{B}_2$ with $\mathbf{W}_1$ and $\mathbf{A}_1$ with $\mathbf{V}_1$. By doing so, we can rewrite Eq. (7) as a sum with indices $\{i_1, s_2, \ldots, s_{D-1}\}$ over products, i.e.,

$$\mathcal{T}[x_1, \ldots, x_D] = \sum_{i_1=1}^{r} \mathbf{V}_1[x_1, i_1] \sum_{s_2=1}^{k} \mathbf{W}_1[i_1, s_2] \mathbf{V}[x_2, s_2] \cdots$$

$$\cdots \sum_{s_{D-2}=1}^{k} \mathbf{W}_{D-3}[s_{D-3}, s_{D-2}] V_{D-2}[x_{D-2}, s_{D-2}] \cdot$$

$$\cdot \sum_{s_{D-1}=1}^{k} \mathbf{W}_{D-2}[s_{D-2}, s_{D-1}] V_{D-1}[x_{D-1}, s_{D-1}] V_D[x_D, s_{D-1}]$$

Fig. B.1 shows an example of such MPS factorization via CP decompositions. We see that we can encode the products over the same indices using a Hadamard product layers, and summations over indices $\{i_1, s_2, \ldots, s_{D-1}\}$ with sum layers parameterized by the $\mathbf{W}_j$, with $1 \leq j < D-1$. More precisely, the sum layers that sum over $s_2$ and $s_{D-1}$ are parameterized by matrices of ones. Each $\mathbf{V}_j$ with $1 \leq j \leq D$ is instead encoded by an input layer depending on the variable $X_j$ and computing $k$ functions $f_l(X_j)$ such that $f_l(x_j) = \mathbf{V}_j[x_j, l]$, with $1 \leq l \leq r$ if $j = 1$ and $1 \leq l \leq k$ if $j > 1$. The tensorized circuit constructed in this way is structured-decomposable, as it is defined on a linear tree RG (e.g., Fig. 2a) induced by the same variable ordering implicitly stated by the MPS factorization (from left to right in Eq. (7), see App. B.4 for details). Fig. B.2 shows the circuit representation corresponding to the MPS reported in Fig. B.1c.

Finally, note that the number of parameters of such tensorized circuit correspond to the size of all $\{\mathbf{W}_j\}_{j=1}^{D-2}$ and $\{\mathbf{V}_j\}_{j=1}^{D}$ introduced above, i.e., overall $\mathcal{O}(D \cdot k^2)$ with $k \leq \min\{r^2, mr\}$. Moreover, the CP decompositions at the beginning can be computed using iterative methods whose iterations require polynomial time (Kolda & Bader, 2009). To retrieve an equivalent BM, we can square the circuit constructed in this way using Alg. 1, which results in a circuit having size $\mathcal{O}(D \cdot k^4)$ (see Prop. B.1). A similar proof can be carried out for showing a reduction of other tensor network structures that can be squared efficiently, such as tree-shaped networks (Cheng et al., 2019).

$\square$

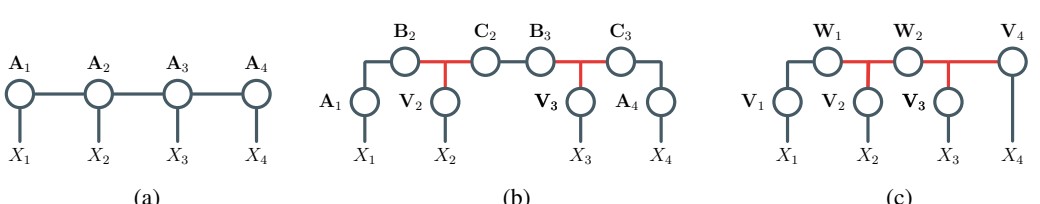

(a)        (b)        (c)

Figure B.1: **Further decomposing a matrix product state (MPS) via CP decompositions.** Tensor networks are represented here using the Penrose graphical notation, where circles denote tensors and their connections denote summations over shared indices, and with variables $X_1, X_2, X_3, X_4$ denoting input indices. Given a MPS (a), we perform a CP decomposition of $\mathbf{A}_2$ and $\mathbf{A}_3$ (b). Red edges denote additional indices given by the CP decompositions. Then, we rename $\mathbf{A}_1$ with $\mathbf{V}_1$, $\mathbf{B}_2$ with $\mathbf{W}_1$. Finally, we contract $\mathbf{C}_2$ with $\mathbf{B}_3$, and $\mathbf{C}_3$ with $\mathbf{A}_4$ resulting in tensors $\mathbf{W}_2$ and $\mathbf{V}_4$, respectively (c). Fig. B.2 shows the tensorized circuit corresponding to such tensor network, where $\mathbf{V}_1, \mathbf{V}_2, \mathbf{V}_3, \mathbf{V}_4$ and $\mathbf{W}_1, \mathbf{W}_2$ parameterize input layers and sum layers, respectively.

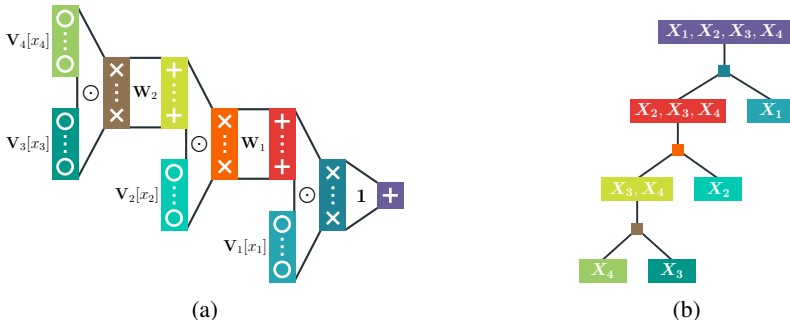

| (a) | (b) |
|---|---|

Figure B.2: **Matrix product states (MPS) as structured-decomposable circuits**. The decomposed MPS over three variables showed in Fig. B.1c can be immediately represented as a tensorized structured-decomposable circuit (a) defined on a linear tree RG (b, matching the colors of layers) having Hadamard product layers and sum layers parameterized by $\mathbf{W}_1, \mathbf{W}_2$ and a row vector of ones $\mathbf{1}$ (for the output sum). Each input layer maps variable assignments $x_1, x_2, x_3, x_4$ to rows in $\mathbf{V}_1, \mathbf{V}_2, \mathbf{V}_3, \mathbf{V}_4$, respectively.

### B.4.1 RELATIONSHIP WITH HIDDEN MARKOV MODELS

MPS tensor networks where each tensor $\mathbf{A}_i$ is non-negative can be seen as inhomogeneous hidden Markov models (HMMs) as showed by Glasser et al. (2019), i.e., where latent state and emitting transitions do not necessarily share parameters. As such, the tensorized structured-decomposable circuit $c$ that is equivalent to a MPS (see App. B.4) is also an inhomogenous HMM if $c$ is monotonic.

In Sec. 5 we experiment with a tensorized monotonic PC that is an inhomogenous HMM to distill a large language model, as to leverage the sequential structure of the sentences. We compare it against a NPC$^2$ that is the squaring of a MPS (also called Born machine (Glasser et al., 2019)) or, equivalently, the squaring of an inhomogenous HMM-like whose parameters can be negative.

### B.5 EXPONENTIAL SEPARATION

**Theorem 1.** There is a class of non-negative functions $\mathcal{F}$ over variables $\mathbf{X}$ that can be compactly represented as shallow squared NMMs (and hence squared non-monotonic PCs) but for which the smallest structured-decomposable monotonic PC computing any $F \in \mathcal{F}$ has size $2^{\Omega(|\mathbf{X}|)}$.

*Proof.* For the proof of Theorem 1, we start by constructing $\mathcal{F}$ by introducing a variant of the *unique disjointness* (UDISJ) problem, which seems to have first been introduced by De Wolf (2003). The variant we consider here is defined over graphs, as detailed in the following definition.

**Definition B.1** (Unique disjointness function). Consider an undirected graph $G = (V, E)$, where $V$ denotes its vertices and $E$ its edges. To every vertex $v \in V$ we associate a Boolean variable $X_v$ and let $\mathbf{X}_V = \{X_v \mid v \in V\}$ be the set of all these variables. The *unique disjointness function* of $G$ is defined as

$$\mathsf{UDISJ}_G(\mathbf{X}_v) := \left(1 - \sum_{uv \in E} X_u X_v\right)^2. \tag{8}$$

**The UDISJ function as a non-monotonic circuit.** We will construct $\mathcal{F}$ as the class of functions $\mathsf{UDISJ}_G$ for graphs $G \in \mathcal{G}$, where $\mathcal{G}$ is a family of graphs that we will choose later. Regardless of the way the class $\mathcal{G}$ is picked, we can compactly represent $\mathsf{UDISJ}_G$ as a squared structured-decomposable (Def. A.3) and non-monotonic circuit as follows. First, we represent the function $c(\mathbf{X}_V) = 1 - \sum_{uv \in E} X_u X_v$ as sum unit computing $1 \cdot a(\mathbf{X}_V) + (-1) \cdot b(\mathbf{X}_V)$ where

- $a$ is a circuit gadget that realizes an unnormalized uniform distribution over the domain of variables in $\mathbf{X}_V$, i.e., $a(\mathbf{X}_V) = \prod_{v \in V}(\mathbb{1}\{X_v = 0\} + \mathbb{1}\{X_v = 1\})$ where $\mathbb{1}\{X_v = 0\}$ (resp. $\mathbb{1}\{X_v = 1\}$) is an indicator function that outputs 1 when $X_v$ is set to 0 (resp. 1);

- $b$ is another sum unit whose inputs are product units over the input units $\mathbb{1}\{X_u = 1\}$, $\mathbb{1}\{X_v = 1\}$ if there is an edge $uv$ in $G$, i.e., $b(\mathbf{X}_V) = \sum_{uv \in E} \mathbb{1}\{X_u = 1\} \cdot \mathbb{1}\{X_v = 1\}$.

Note that $b$ may not be smooth, but we can easily smooth it by adding to every product an additional input that is a circuit similar to $a$ that outputs 1 for any input $\mathbf{X}_{\overline{uv}}$, where $\mathbf{X}_{\overline{uv}} = \mathbf{X}_V \setminus \{X_u, X_v\}$. Since $c$ is structured-decomposable (Def. A.3), we can easily multiply it with itself to realize $c^2$ that would be still a structured-decomposable circuit whose size is polynomially bounded as $|c^2| \in \mathcal{O}(|c|^2)$ (Vergari et al., 2021). Note that in this case we have that $|c|$ is a polynomial in the number of variables (or vertices) $|\mathbf{X}_V|$ by the construction above. Furthermore, note that $c^2$ is non-monotonic as one of its sum unit has negative parameters (i.e., $-1$) to encode the subtraction in Eq. (8).

**The lower bound for monotonic circuits.** To prove the exponential lower bound for monotonic circuits in Theorem 1, we will use an approach that has been used in several other works (Martens & Medabalimi, 2014; de Colnet & Mengel, 2021). This approach is based on representing a decomposable circuit (and hence a structured-decomposable one) as a shallow mixture whose components are *balanced products*, as formalized next.

**Definition B.2** (Balanced decomposable product). Let $\mathbf{X}$ be a set of variables. A *balanced decomposable product* over $\mathbf{X}$ is a function from $\mathbf{X}$ to $\mathbb{R}$ that can be written as $f(\mathbf{Y}) \times h(\mathbf{Z})$ where $f$ and $h$ are functions to $\mathbb{R}$, $\times$ is an alias for the product (when used to multiply functions from now on), and $(\mathbf{Y}, \mathbf{Z})$ is a *balanced* partitioning of $\mathbf{X}$, i.e., $\mathbf{Y} \cup \mathbf{Z} = \mathbf{X}$, $\mathbf{Y} \cap \mathbf{Z} = \varnothing$ with $|\mathbf{X}|/3 \leq |\mathbf{Y}|, |\mathbf{Z}| \leq 2|\mathbf{X}|/3$.

**Theorem B.1** (Martens & Medabalimi (2014)). Let $F$ be a non-negative function over Boolean variables $\mathbf{X}$ computed by a smooth and decomposable circuit $c$. Then, $F$ can be written as a sum of $N$ balanced decomposable products (Def. B.2) over $\mathbf{X}$, with $N \leq |c|$ in the form[4]

$$F(\mathbf{X}) = \sum_{k=1}^{N} f_k(\mathbf{Y}_k) \times h_k(\mathbf{Z}_k),$$

where $(\mathbf{Y}_k, \mathbf{Z}_k)$ is balanced a partitioning of $\mathbf{X}$ for $1 \leq k \leq N$. If $c$ is structured-decomposable, the $N$ partitions $\{(\mathbf{Y}_k, \mathbf{Z}_k)\}_{k=1}^{N}$ are all identical. Moreover, if $c$ is monotonic, then all $f_k, h_k$ only compute non-negative values.

Intuitively, Thm. B.1 tells us that to lower bound the size of $c$ we can lower bound $N$. To this end, we first encode the UDISJ function (Eq. (8)) as a sum of $N$ balanced products and show the exponential growth of $N$ for a particular family of graphs. We start with a special case for a representation in the following proposition.

**Proposition B.2.** Let $G_n$ be a matching of size $n$, i.e., a graph consisting of $n$ edges none of which share any vertices. Assume that the UDISJ function (Eq. (8)) for $G_n$ is written as a sum of products of balanced partitions

$$\mathsf{UDISJ}_{G_n}(\mathbf{Y}, \mathbf{Z}) = \sum_{k=1}^{N} f_k(\mathbf{Y}) \times h_k(\mathbf{Z}),$$

where for every edge $uv$ in $G_n$ we have that $X_u \in \mathbf{Y}$ and $X_v \in \mathbf{Z}$, and $f_k, h_k$ are non-negative functions. Then $N = 2^{\Omega(n)}$.

To prove the above result, we will make an argument on the rank of the so-called *communication matrix*, also known as the *value matrix*, for a function $F$ and a fixed partition $(\mathbf{Y}, \mathbf{Z})$.

**Definition B.3** (Communication matrix, or value matrix (de Colnet & Mengel, 2021)). Let $F$ be a function over $(\mathbf{Y}, \mathbf{Z})$, its communication matrix $M_F$ is a $2^{|\mathbf{Y}|} \times 2^{|\mathbf{Z}|}$ matrix whose rows (resp. columns) are uniquely indexed by assignments to $\mathbf{Y}$ (resp. $\mathbf{Z}$) such that for a pair of index[5] $(i_{\mathbf{Y}}, j_{\mathbf{Z}})$, the entry at the row $i_{\mathbf{Y}}$ and column $j_{\mathbf{Z}}$ in $M_F$ is $F(i_{\mathbf{Y}}, j_{\mathbf{Z}})$.

---

[4]In Martens & Medabalimi (2014), Theorem 38, this result is stated with $N \leq |c|^2$. The square materializes from the fact that they reduce their circuits to have all their inner units to have exactly two inputs, as we already assume, following de Colnet & Mengel (2021).

[5]An index $i_{\mathbf{Y}}$ (resp. $j_{\mathbf{Z}}$) is a complete assignment to Boolean variables in $\mathbf{Y}$ (resp. $\mathbf{Z}$). See Example 1.

**Example 1.** *Let us consider a simple matching on 6 vertices, where* **Y** *correspond to the first 3 vertices, and* **Z** *to the last 3, and where there is an edge between the first, second and third vertices of* **Y** *and* **Z***. The matrix* $M_F$ *is an 8-by-8 matrix, a row and a column for each assignment of the 3 binary variables associated to each vertex; it is given by*

| **Y**\\**Z** | *000* | *100* | *010* | *001* | *110* | *101* | *011* | *111* |
|---|---|---|---|---|---|---|---|---|
| *000* | *1* | *1* | *1* | *1* | *1* | *1* | *1* | *1* |
| *100* | *1* | *0* | *1* | *1* | *0* | *0* | *1* | *0* |
| *010* | *1* | *1* | *0* | *1* | *0* | *1* | *0* | *0* |
| *001* | *1* | *1* | *1* | *0* | *1* | *0* | *0* | *0* |
| *110* | *1* | *0* | *0* | *1* | *1* | *0* | *0* | *1* |
| *101* | *1* | *0* | *1* | *0* | *0* | *1* | *0* | *1* |
| *011* | *1* | *1* | *0* | *0* | *0* | *0* | *1* | *1* |
| *111* | *1* | *0* | *0* | *0* | *1* | *1* | *1* | *4* |

Note that the name UDISJ comes from the fact that $M_F(i,j) = 0$ if and only if **Y** and **Z** share a single entry equal to 1. In the following, we will rely on the following quantity.

**Definition B.4** (Non-negative rank). The non-negative rank of a non-negative matrix $A \in \mathbb{R}_+^{m \times n}$, denoted $\mathsf{rank}_+(A)$, is the smallest $k$ such that there exist $k$ nonnegative rank-one matrices $\{A_i\}_{i=1}^k$ such that $A = \sum_{i=1}^k A_i$. Equivalently, it is the smallest $k$ such that there exists two non-negative matrices $B \in \mathbb{R}_+^{m \times k}$ and $C \in \mathbb{R}_+^{k \times n}$ such that $A = BC$.

Given a function $F$ written as a sum over $N$ decomposable products (see Thm. B.1) over a fixed partition $(\mathbf{Y}, \mathbf{Z})$, we now show that the non-negative rank of its communication matrix $M_F$ (Def. B.3) is a lower bound of $N$.

**Lemma B.1.** Let $F(\mathbf{X}) = \sum_{k=1}^N f_k(\mathbf{Y}) \times h_k(\mathbf{Z})$ where $f_k$ and $h_k$ are non-negative functions and let $M_F$ be the communication matrix (Def. B.3) of $F$ for the partition $(\mathbf{Y}, \mathbf{Z})$, then it holds that

$$\mathsf{rank}_+(M_F) \leq N.$$

*Proof.* This proof is an easy extension of the proof of Lemma 13 from de Colnet & Mengel (2021). Assume w.l.o.g. that $f_k(\mathbf{Y}) \times h_k(\mathbf{Z}) \neq 0$ for any complete assignment to **Y** and **Z**.[6] Let $M_k$ denote the communication matrix of the function $f_k(\mathbf{Y}) \times h_k(\mathbf{Z})$. By construction, we have that $M_F = \sum_{k=1}^N M_k$. Furthermore, since all values in $M_F$ are non-negative by definition, $\mathsf{rank}_+(M_k)$ is defined for all $k$ and by sub-additivity of the non-negative rank we have that $\mathsf{rank}_+(M_F) \leq \sum_{k=1}^N \mathsf{rank}_+(M_k)$. To conclude the proof, it is sufficient to show that $M_k$ are rank-1 matrices, i.e., $\mathsf{rank}_+(M_k) = 1$. To this end, consider an arbitrary $k$. Since $f_k(\mathbf{Y}) \times h_k(\mathbf{Z}) \neq 0$, there is a row in $M_k$ that is not a row of zeros. Say it is indexed by $i_{\mathbf{Y}}$, then its entries are of the form $f_k(i_{\mathbf{Y}}) \times h_k(j_{\mathbf{Z}})$ for varying $j_{\mathbf{Z}}$. In any other rows indexed by $i'_{\mathbf{Y}}$ we have $f_k(i'_{\mathbf{Y}}) \times h_k(j_{\mathbf{Z}}) = (f_k(i'_{\mathbf{Y}})/f_k(i_{\mathbf{Y}})) \times f_k(i_{\mathbf{Y}}) \times h_k(j_{\mathbf{Z}})$ for varying $j_{\mathbf{Z}}$. Consequently, all rows are non-negative multiples of the $i_{\mathbf{Y}}$ row, and therefore $\mathsf{rank}_+(M_k) = 1$. □

To complete the proof of Prop. B.2, we leverage a known lower bound of the non-negative rank of the communication matrix of the UDISJ problem. The interested reader can find more information on this result in the books Roughgarden (2016), Gillis (2020) and the references therein.

**Theorem B.2** (Fiorini et al. (2015)). Let a UDISJ function defined as in Prop. B.2, and $M_{\mathsf{UDISJ}}$ be its communication matrix over a partition $(\mathbf{Y}, \mathbf{Z})$, then it holds that

$$(3/2)^n \leq \mathsf{rank}_+(M_{\mathsf{UDISJ}}).$$

Using Thm. B.2 and Lem. B.1, we directly get Prop. B.2. So we have shown that, for a fixed partition of variables $(\mathbf{Y}, \mathbf{Z})$, every monotonic circuit $c$ encoding the UDISJ function (Eq. (8)) of a matching of size $n$ has size $|c| \geq 2^{\Omega(n)}$. However, the smallest non-monotonic circuit encoding the same function has polynomial size in $n$ (see the construction of the UDISJ function as a circuit above). Now, to complete the proof for the exponential lower bound in Theorem 1, we need to find

---

[6]If this were not the case we could simply drop the term from the summation, which would clearly reduce the number of summands.

a function class $\mathcal{F}$ where this result holds for all possible partitions $(\mathbf{Y}, \mathbf{Z})$. Such function class consists of UDISJ functions over a family of graphs, as detailed in the following proposition.

**Proposition B.3.** There is a family of graphs $\mathcal{G}$ such that for every graph $G_n = (V_n, E_n) \in \mathcal{G}$ we have $|V_n| = |E_n| = \mathcal{O}(n)$, and any monotonic structured-decomposable circuit representation of $\mathsf{UDISJ}_{G_n}$ has size $2^{\Omega(n)}$.

*Proof.* We prove it by constructing a class of so-called *expander graphs*, which we introduce next. We say that a graph $G = (V, E)$ has expansion $\varepsilon$ if, for every subset $V'$ of $V$ of size at most $|V|/2$, there are at least $\varepsilon|V'|$ edges from $V'$ to $V \setminus V'$ in $G$. It is well-known, see e.g. Hoory et al. (2006), that there are constants $\varepsilon > 0$ and $d \in \mathbb{N}$ and a family $(G_n)_{n \in \mathbb{N}}$ of graphs such that $G_n$ has at least $n$ vertices, expansion $\varepsilon$ and maximal degree $d$. We fix such a family of graphs in the remainder and denote by $V_n$ (resp. $E_n$) the vertex set (resp. the edge set) of $G_n$.

Let $c$ be a monotonic structured-decomposable circuit of size $N$ computing $\mathsf{UDISJ}_{G_n}$. Then, by using Thm. B.1, we can write it as

$$\mathsf{UDISJ}_{G_n}(\mathbf{Y}, \mathbf{Z}) = \sum_{k=1}^{N} f_k(\mathbf{Y}) \times h_k(\mathbf{Z}) \tag{9}$$

where $(\mathbf{Y}, \mathbf{Z})$ is a balanced partition of $\mathbf{X}_V$. Let $V_{\mathbf{Y}} = \{v \in V_n \mid X_v \in \mathbf{Y}\}$ and $V_{\mathbf{Z}} = \{v \in V_n \mid X_v \in \mathbf{Z}\}$. Then $(V_{\mathbf{Y}}, V_{\mathbf{Z}})$ form a balanced partition of $V_n$. By the expansion of $G_n$, it follows that there are $\Omega(n)$ edges from vertices in $V_{\mathbf{Y}}$ to vertices in $V_{\mathbf{Z}}$. By greedily choosing some of those edges and using the bounded degree of $G_n$, we can construct an edge set $E'_n$ of size $\Omega(n)$ that is a matching between $\mathbf{Y}$ and $\mathbf{Z}$, i.e., all edges in $E'_n$ go from $\mathbf{Y}$ to $\mathbf{Z}$ and every vertex in $V_n$ is incident to only one edge in $E'_n$. Let $V'_n$ be the set of endpoints in $E'_n$ and $\mathbf{X}_{V'} \subseteq \mathbf{X}_V$ be the variables associated to them. We construct a new circuit $c'$ from $c$ by substituting all input units for variables $X_v$ that are not in $\mathbf{X}_{V'_n}$ by 0. Clearly, $|c'| \leq |c|$ and hence all the lower bounds for $|c'|$ are lower bounds for $|c|$. Let $\overline{\mathbf{Y}} = \mathbf{X}_{V'_n} \cap \mathbf{Y}$ and $\overline{\mathbf{Z}} = \mathbf{X}_{V'_n} \cap \mathbf{Z}$. By construction $c'$ computes the function

$$\mathsf{UDISJ}_{G'_n}(\overline{\mathbf{Y}}, \overline{\mathbf{Z}}) = \left(1 - \sum_{uv \in E'_n} X_u X_v\right)^2$$

which corresponds to solving the UDISJ problem over the graph $G'_n = (V'_n, E'_n)$. From Eq. (9) recover that

$$\mathsf{UDISJ}_{G'_n}(\overline{\mathbf{Y}}, \overline{\mathbf{Z}}) = \sum_{k=1}^{N} f'_k(\overline{\mathbf{Y}}) \times h'_k(\overline{\mathbf{Z}}),$$

where $f'_k$ (resp. $h'_k$) are obtained from $f_k$ (resp. $h_k$) by setting all the variables not in $\mathbf{X}_{V'_n}$ to 0. Since $c'$ is monotonic by construction and $|E'_n| = \Omega(n)$, from Prop. B.2 it follows that $N = 2^{\Omega(n)}$. $\square$

Prop. B.3 concludes the proof of Theorem 1, as we showed the existence of family of graphs for which the smallest structured-decomposable monotonic circuit computing the UDISJ function over $n$ variables has size $2^{\Omega(n)}$. However, the smallest structured-decomposable non-monotonic circuit has size polynomial in $n$, whose construction has been detailed at the beginning of our proof. $\square$

### B.6 SQUARING DETERMINISTIC CIRCUITS

In Sec. 4.1 we argued that squaring any non-monotonic, smooth, decomposable (Def. A.2), and deterministic (Def. A.5) circuit yields a monotonic and deterministic PC. As a consequence, any function computed by a NPC$^2$ that is deterministic can be computed by a monotonic and deterministic PC. Therefore, we are interested in squaring structured-decomposable circuits that are *not* deterministic. Below we formally prove Proposition 4.

**Proposition 4.** Let $c$ be a smooth, decomposable and deterministic circuit over variables $\mathbf{X}$ possibly computing a negative function. Then, the squared circuit $c^2$ is monotonic and has the same structure (hence size) of $c$.

*Proof.* The proof is by induction. Let $n \in c$ be a product unit that computes $c_n(\mathbf{Z}) = \prod_{i \in \mathsf{in}(n)} c_n(\mathbf{Z}_i)$, with $\mathbf{Z} \subseteq \mathbf{X}$ and $(\mathbf{Z}_1, \dots, \mathbf{Z}_{|\mathsf{in}(n)|})$ forming a partitioning of $\mathbf{Z}$. Then its squaring computes $c_n^2(\mathbf{Z}) = \prod_{i \in \mathsf{in}(n)} c_n^2(\mathbf{Z}_i)$. Now consider a sum unit $n \in c$ that computes $c_n(\mathbf{Z}) = \sum_{i \in \mathsf{in}(n)} w_i c_i(\mathbf{Z})$ with $\mathbf{Z} \subseteq \mathbf{X}$ and $w_i \in \mathbb{R}$. Then its squaring computes $c_n^2(\mathbf{Z}) = \sum_{i \in \mathsf{in}(n)} \sum_{j \in \mathsf{in}(n)} w_i w_j c_i(\mathbf{Z}) c_j(\mathbf{Z})$. Since $c$ is deterministic (Def. A.5), for any $i, j$ with $i \neq j$ either $c_i(\mathbf{Z})$ or $c_j(\mathbf{Z})$ is zero for any assignment to $\mathbf{Z}$. Therefore, we have that

$$c_n^2(\mathbf{Z}) = \sum_{i \in \mathsf{in}(n)} w^2 c_i^2(\mathbf{Z}). \tag{10}$$

This implies that in deterministic circuits, squaring does not introduce additional components that encode (possibly negative) cross-products. The base case is defined on an input unit $n$ that models a function $f_n$, and hence its squaring is an input unit that models $f_n^2$. By induction $c^2$ is constructed from $c$ by squaring the parameters of sum units $w_i$ and squaring the functions $f_n$ modeled by input units. Moreover, the number of inputs of each sum unit remains the same, as we observe in Eq. (10), and thus $c^2$ and $c$ have the same size. □

## C EFFICIENT LEARNING OF NPC$^2$S

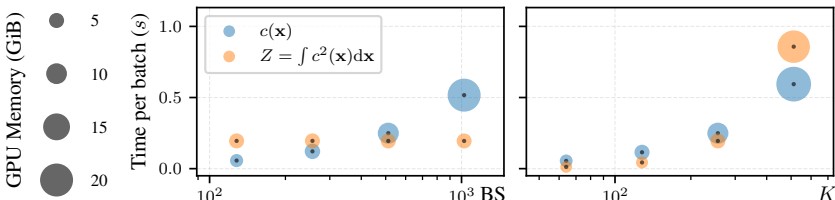

Figure C.1: **Evaluating the squared circuit representation adds little overhead during training.** By learning by MLE (Eq. (4)) and batched gradient descent, the time and space required to compute the partition function $Z$ of $c^2$ is constant w.r.t. the batch size (BS) (left). By fixing the batch size to 512 and varying the output dimensionality ($K$) of each layer (right), the resources needed to compute $Z$ are similar to the ones needed to evaluate $c$ (i.e., $c(\mathbf{X})$). For the left figure, we fix $K = 256$ and vary the BS, while for the right figure we fix BS $= 512$ and vary $K$. The plots share the y-axis.

In this section, we investigate the computational cost of learning NPC$^2$s with a series of benchmarks, showing that NPC$^2$s add little computational overhead over traditional monotonic PCs (MPCs).

**Efficient renormalization in practice.** As suggested by the MLE objective (Eq. (4)), squaring the tensorized circuit $c$ with Alg. 1 is only required to compute the partition function $Z = \int c^2(\mathbf{x}) d\mathbf{x}$. In addition, we need to compute $Z$ only once per parameter update via gradient ascent, as $Z$ does not depend on the training data. For these reasons, the increased computational burden of evaluating a squared circuit (see Proposition 1) as to compute $Z$ is negligible, and it is independent w.r.t. the batch size. Fig. C.1 illustrates this aspect by comparing the time needed to evaluate $c$ on a batch of data and to compute the partition function $Z$. The results showed in Fig. C.1 are obtained by running benchmarks on NPC$^2$s that are similar in size to the ones we experiment with in Sec. 5. That is, we benchmark a mixture of 32 NPC$^2$s, each having an architecture built from a randomly-generated tree RG (see App. F for details) approximating the density function of BSDS300 (the data set with highest number of variables, see Table H.1). The input layers compute Gaussian distributions.

**Training efficiency on UCI data sets.** We benchmark the computational cost of learning NPC$^2$s on UCI data sets (Table H.1). Fig. C.2 compares time and memory required to learn the best NPC$^2$s and MPCs showed in Fig. 4, while Fig. C.3 compares time and memory required to learn them in a worse scenario for NPC$^2$s where the batch size is small and the layer dimensionality is large, as NPC$^2$s benefit from using large batch sizes as discussed above. NPC$^2$s add very little overhead during training in most configurations when compared to MPCs, as computing the partition function $Z$ is comparable to evaluating MPCs on a batch of samples. In particular, on Gas ($|\mathbf{X}| = 8$), NPC$^2$ takes more time and memory to compute $Z$ (times are 6ms and 121ms, while memory allocations are 0.6GiB and 5.8GiB), but it is only slightly more than the cost of computing $c$ for MPCs (time

144ms and memory 4.4GiB). Moreover, note that NPC$^2$s achieve about a $\times 2$ improvement on the log-likelihood on Gas. On the much higher dimensional data set BSDS300 ($|\mathbf{X}| = 63$) instead, we found that training NPC$^2$ is even cheaper as it requires fewer parameters while still achieving an higher log-likelihood (128.38 rather than 123.3).

**Hardware and significance of benchmarks.** The benchmarks mentioned above and illustrated in Figs. C.1 to C.3 have been run on a single NVIDIA RTX A6000 with 48GiB of memory. The measured times are averaged over 50 independent circuit evaluations.

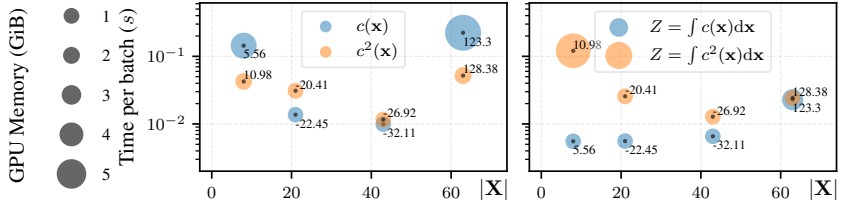

Figure C.2: **NPC$^2$s add little overhead during training on real-world data sets**, while improving log-likelihoods. We evaluate time and memory required by monotonic PCs (MPCs) and NPC$^2$s to perform one optimization step on UCI data sets (Gas, Hepmass, MiniBooNE, BSDS300) with number of variables $|\mathbf{X}|$ and using the best hyperparameters found (see App. H.3). We benchmark the computation of $c(\mathbf{x})$ by MPCs and $c^2(\mathbf{x})$ by NPC$^2$s on a batch $\mathbf{x}$ of data (left), as well as the partition functions $Z$ for both models (right), and label the data points with the final log-likelihoods achieved by the corresponding models (as also reported in Fig. 4). The plots share the y-axis. For NPC$^2$s, computing the partition function $Z$ is more expensive both in time and memory (right), but it is still very similar to the cost of evaluating $c(\mathbf{x})$ or $c^2(\mathbf{x})$ (left).

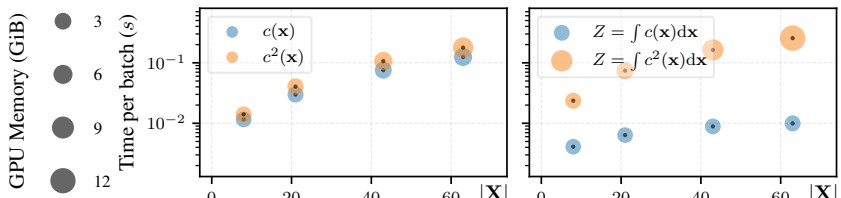

Figure C.3: **NPC$^2$s add little overhead during training even with relatively small batch sizes.** We evaluate time and memory required by monotonic PCs (MPCs) and NPC$^2$s to perform one optimization step on UCI data sets (Gas, Hepmass, MiniBooNE, BSDS300) with respect to the number of variables $|\mathbf{X}|$ and using the same hyperparameters (512 as batch size, 512 as layer dimensionality, and Gaussian input layers). The plots share the y-axis. The cost of computing $c^2(\mathbf{x})$ on a batch $\mathbf{x}$ of data by NPC$^2$s is only slightly higher than the cost of computing $c(\mathbf{x})$ by MPCs (left), while the cost of computing $Z$ for NPC$^2$s is comparable to evaluating $c^2(\mathbf{x})$ or $c(\mathbf{x})$ (right).

## D  THE SIGNED LOG-SUM-EXP TRICK

Scaling squared non-monotonic PCs to more than a few tens (resp. hundreds) of variables without performing computations in log-space is infeasible in 32-bit (resp. 64-bit) floating point arithmetic, as we illustrate in Fig. D.1. For this reason, we *must* perform computations in the log-space even in presence of negative values. The idea is to represent non-zero outputs $\mathbf{y} \in \mathbb{R}^S$ of each layer in terms of the element-wise logarithm of their absolute value $\log |\mathbf{y}|$ and their element-wise sign $\mathrm{sign}(\mathbf{y}) \in \{-1, 1\}^S$, i.e., such that $\mathbf{y} = \mathrm{sign}(\mathbf{y}) \odot \exp(\log |\mathbf{y}|)$.

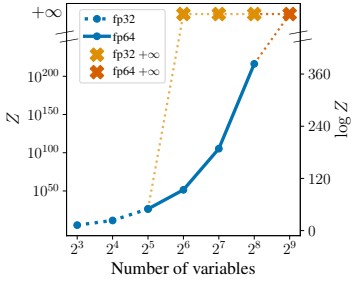

Figure D.1: **Squared non-monotonic PCs cannot scale without performing computations in log-space.** Partition functions (and their *natural* logarithm) of squared non-monotonic PCs having Gaussian input units, with increasing number of variables $V$ and having depth $\lceil \log_2 V \rceil$ computed using 32-bit and 64-bit floating point arithmetic.

In practice, we evaluate product and sum layers according to the following evaluation rules. Given an Hadamard product layer $\ell$, then it computes and propagates both $\log |\ell| = \sum_{i=1}^{N} \log |\ell_i|$ and $\operatorname{sign}(\ell) = \bigodot_{i=1}^{N} \operatorname{sign}(\ell_i)$ for some inputs $\{\ell_i\}_{i=1}^{N}$. Given a sum layer $\ell$ parameterized by $\mathbf{W} \in \mathbb{R}^{S \times K}$ and having $\ell'$ as input layer, then it computes and propagates both $\log |\ell| = \boldsymbol{\alpha} + \log |\mathbf{s}|$ and $\operatorname{sign}(\ell) = \operatorname{sign}(\mathbf{s})$ where $\boldsymbol{\alpha}$ and $\mathbf{s}$ are defined as

$$\boldsymbol{\alpha} = \mathbf{1} \cdot \max_{1 \leq j \leq S} \{\log |\ell'[j]|\} \qquad \mathbf{s} = \mathbf{W} \left( \operatorname{sign}(\ell') \odot \exp(\log |\ell'| - \boldsymbol{\alpha}) \right)$$

by assuming $\mathbf{s} \neq \mathbf{0}$, $\mathbf{1}$ denoting a $S$-dimensional vector of ones, $\ell'[j]$ denoting the $j$-th entry of the output of $\ell'$, and $\exp$ being applied element-wise. We call *signed log-sum-exp trick* the evaluation rule above for sum layers, which generalizes the log-sum-exp trick (Blanchard et al., 2021) that is used to evaluate tensorized monotonic PC architectures (Peharz et al., 2020a).

For the more general definition of tensorized circuits instead (Def. A.6), given a Kronecker product layer $\ell$, then it computes both $\log |\ell| = \bigoplus_{i=1}^{N} \log |\ell_i|$ and $\operatorname{sign}(\ell) = \bigotimes_{i=1}^{N} \operatorname{sign}(\ell_i)$, where $\bigoplus$ denotes an operator similar to the Kronecker product but computing sums rather than products.

## E   SPLINES AS EXPRESSIVE INPUT COMPONENTS

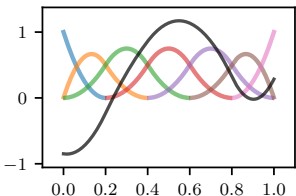

Figure E.1: **Splines represent a class of flexible non-linear functions.** A quadratic ($k = 2$) spline (in black) over $n = 4$ knots chosen uniformly in $(0, 1)$ (i.e, 0.2, 0.4, 0.6 and 0.8) is computed by a linear combination of $n + k + 1 = 7$ distinct basis functions (each colored differently).

Polynomials defined on fixed intervals are candidate functions to be modeled by components (resp. input layers) of squared NMMs (Sec. 2) (resp. NPC$^2$s Sec. 3). This is because they can be negative function and their product can be tractably integrated. In particular, we experiment with piecewise polynomials, also called *splines*. An univariate spline function of order $k$ is a piecewise polynomial defined on a variable $X$, and the $n$ values of $X$ where polynomials meet are called *knots*. *B-splines* of order $k$ are basis functions for continuous spline functions of the same degree. In practice, we can represent any spline function $f$ of order $k$ defined over $n$ knots inside an interval $(a, b)$ as a linear combination of $n + k + 1$ basis functions, i.e.,

$$f(X) = \sum_{i=1}^{n+k+1} \alpha_i B_{i,k}(X) \tag{11}$$

where $\alpha_i \in \mathbb{R}$ are the parameters of the spline and $B_{i,k}(X)$ are polynomials of order $k$ (i.e., the basis of $f$), which are unequivocally determined by the choice of the $n$ knots. In particular, each $B_{i,k}(X)$ is a non-negative polynomial that is recursively defined with the Cox-de-Boor formula (de Boor, 1971; Piegl & Tiller, 1995). Given two splines $f, g$ of order $k$ defined over $n$ knots and represented in terms of $n + k + 1$ basis functions as in Eq. (11), we can write their product integral as

$$\int_a^b f(X)g(X) \,\mathrm{d}X = \sum_{i=1}^{n+k+1} \sum_{j=1}^{n+k+1} \alpha_i \beta_j \int_a^b B_{i,k}(X)B_{j,k}(X) \,\mathrm{d}X \tag{12}$$

where $\alpha_i \in \mathbb{R}$ (resp. $\beta_j \in \mathbb{R}$) denote the parameters of $f$ (resp. $g$). Therefore, integrating a product of splines requires integrating products of their basis functions. Among the various way of computing Eq. (12) exactly (Vermeulen et al., 1992), we can do it in time $\mathcal{O}(n^2 \cdot k^2)$ by representing the product $B_{i,k}(X)B_{j,k}(X)$ as the basis polynomial of another B-spline of order $2k+1$, and finally integrating it in the interval of definition. Fig. E.1 shows an example of a spline.

Since each $B_{i,k}$ is non-negative, we can use B-splines as components (resp. modeled by input layers) of traditional MMs (resp. monotonic PCs) by assuming each spline parameter $\alpha_i$ to be non-negative. This is the case of monotonic PCs we experimented with in Sec. 5, where non-negativity is guaranteed via exponentiation of the parameters.

## F  TREE REGION GRAPHS

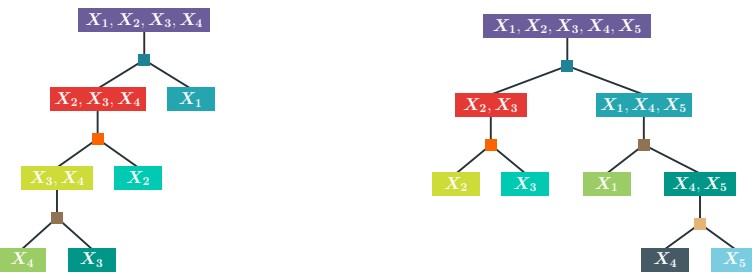

Figure F.1: **Different ways to construct region graphs.** The left figure illustrates a linear tree (LT) region graph (Def. 2) over four variables, which decomposes variables one by one. The right figure shows a possible binary tree (RT) region graph over five variables, which recursively splits them.

Since we require structured-decomposability to square circuits (see Sec. 3.2), we construct their architecture based on tree RGs (Def. 2). We choose to experiment with two kinds of tree RGs: *binary tree* (BT) and *linear tree* (LT). Following Peharz et al. (2020b), the BT is built by recursively partitioning variables evenly and randomly until regions with only one variable are obtained. The LT is built by (1) shuffling the variables randomly and then (2) recursively partitioning variables one by one, i.e., a set of variables $\{X_i, \ldots, X_D\}$ is partitioned in $\{X_i\}$ and $\{X_{i+1}, \ldots, X_D\}$ for $1 \le i \le D - 1$. Fig. F.1 shows examples of LT and BT RGs. Note that the LT is the same on which the circuit representation of matrix-product states (MPS) (Pérez-García et al., 2007) and TTDE (Novikov et al., 2021) depend on (see also Sec. 4 and App. B.4).

## G  ADDITIONAL RELATED WORKS

**Squared neural family (SNEFY)** (Tsuchida et al., 2023) have been concurrently proposed as a class of models squaring the 2-norm of the output of a single-hidden-layer neural network. Under certain parametric conditions, SNEFYs can be re-normalized as to model a density function, but they do not guarantee tractable marginalization of any subset of variables as our NPC$^2$s do, unless they encode a fully-factorized distribution, which would limit their expressiveness. Hence, SNEFYs can be employed in our NPC$^2$s to model multivariate units in input layers with bounded scopes.

The **rich literature of PCs** provides several algorithms to learn both the structure and the parameters of circuits (Poon & Domingos, 2011; Peharz et al., 2017; Di Mauro et al., 2021; Dang et al., 2021; Liu & Van den Broeck, 2021; Liu et al., 2023). However, in these works circuits are always assumed to be monotonic. A first work considering subtractions is Dennis (2016) which generalizes the ad-hoc constraints over Gaussian NMMs (Zhang & Zhang, 2005) to deep PCs over Gaussian inputs by constraining their structure and reparameterizing their sum weights. Shallow NMM represented as squared circuits have been investigated for low-dimensional categorical distributions in Loconte et al. (2023). Concurrently, Sladek et al. (2023) investigated interleaving PSD models and PCs to represent deep NMMs in low-dimensional settings. The resulting model can be interpreted as a sum of squared circuits. Circuit representations encoding probability generating functions allow negative coefficients in symbolic computational graphs (Zhang et al., 2021), differently from them

we encode probability densities and masses. Non-monotonic PCs have been recently proven to be able to compactly represent determinantal point processes, for which no compact monotonic circuit representation exists (Broadrick et al., 2024).

The **relationship with tensor networks** and PCs has been hinted in (Glasser et al., 2019) and (Novikov et al., 2021) for matrix-product states. However, they did neither provide a formal reduction nor a highlighted the structural properties needed to tractably perform the squaring and marginalize variables. Glasser et al. (2019) showed a number of bounds over ranks of matrix-product states and Born machines, however they do not provide a separation between $NPC^2$s and monotonic PCs as we do, nor their results generalize to any region graph, including tree-shaped networks as our Theorem 1 does for structured-decomposable monotonic PCs. Born machines and squared tree-shaped tensor networks have been explored for distribution estimation (Han et al., 2018; Cheng et al., 2019) but, differently from our $NPC^2$s, they were not equipped with non-linearities at the inputs and their evaluation was limited to small scale binary data. Bailly (2011) proposed squared probabilistic automata that are similar to Born machines but supporting inputs of any length by sharing the same parameter tensors across steps. By applying the construction used to show Proposition 3, we can represent such models as $NPC^2$s where the parameters of sum and input layers are shared. Jaini et al. (2018) draw a connection between monotonic PCs, latent tree models (Choi et al., 2011) and hierarchical tensor mixture models (Hackbusch, 2012) with non-negative parameters, showing an exponential separation between shallow and deep monotonic circuits.

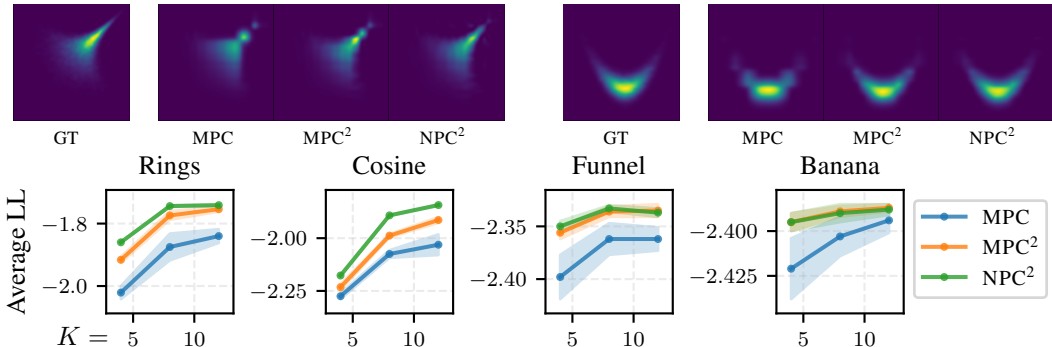

Figure H.1: **Negative parameters increases the expressiveness of NPC²s.** From left to right (above) and for each bivariate density, we show the ground truth (GT) and its estimation by a monotonic PC (MPC), a squared monotonic PC (MPC²), and a NPC² having input layers computing quadratic splines (App. E) and with the same number of parameters. Moreover, (below) we show the average log-likelihoods (and one standard deviation with 10 independent runs) on unseen data achieved by a monotonic MPC, a squared monotonic MPC², and a NPC² by increasing the dimensionality of input layers $K$.

## H  EXPERIMENTAL SETTINGS AND COMPLEMENTARY RESULTS

### H.1  CONTINUOUS SYNTHETIC DATA

Following (Wenliang et al., 2019) we experiment with monotonic PCs, their squaring and NPC²s on synthetic continuous 2D data sets, named *rings*, *cosine*, *funnel* and *banana*. We generate each synthetic data set by sampling $10\_000/1\_000/2\_000$ training/validation/test samples. In these experiments, we are interested in studying whether NPC²s can be more expressive in practice, without making assumptions on the data distribution and therefore choosing parametric distributions as components. For this reason, we choose components computing the product of univariate spline functions (see App. E) over 32 knots that are uniformly chosen in the data domain. In particular, for monotonic mixtures we restrict the spline coefficients to be non-negative.

**Learning and hyperparameters**. Since the data is bivariate, the tree on which PCs are defined on consists of just one region that is split in half. All models are learned by batched stochastic gradient descent using the Adam optimizer with default learning rate (Kingma & Ba, 2015) and a batch size of 256. The parameters of all mixtures are initialized by sampling uniformly between 0 and 1. Furthermore, monotonicity in (squared) PCs is ensured by exponentiating the parameters.

Fig. 3 shows the density functions estimated from data sets *rings* and *cosine*, when using 8 and 12 components, respectively. Moreover, Fig. H.1 report the log-likelihoods and other density functions learned from data sets *funnel* and *banana*, when using 4 components.

### H.2  DISCRETE SYNTHETIC DATA

For our experiments investigating the flexibility of input layers of NPC²s (Sec. 2) in case of discrete data (Sec. 5), we quantize the bivariate continuous synthetic data sets reported in App. H.1. That is, we discretize both continuous variables using 32 uniform bins each. The resulting target distribution is therefore a probability mass function over two finitely discrete variables.

We experiment with monotonic PCs, their squaring and NPC²s with two families of input layers. First, we investigate very flexible input layers for finitely discrete data: categoricals for monotonic PCs and embeddings for NPC²s. Second, we experiment with the less flexible but more parameter-efficient Binomials. The learning and hyperparameters setting are the same used for the continuous data (see App. H.1). Fig. H.2 shows that there is little advantage in subtracting probability mass with respect to monotonic PCs having categorical components. However, in case of the less flexible Binomial components, NPC²s capture the target distribution significantly better. This is also confirmed by the log-likelihoods on unseen data, which we show in Fig. H.2.

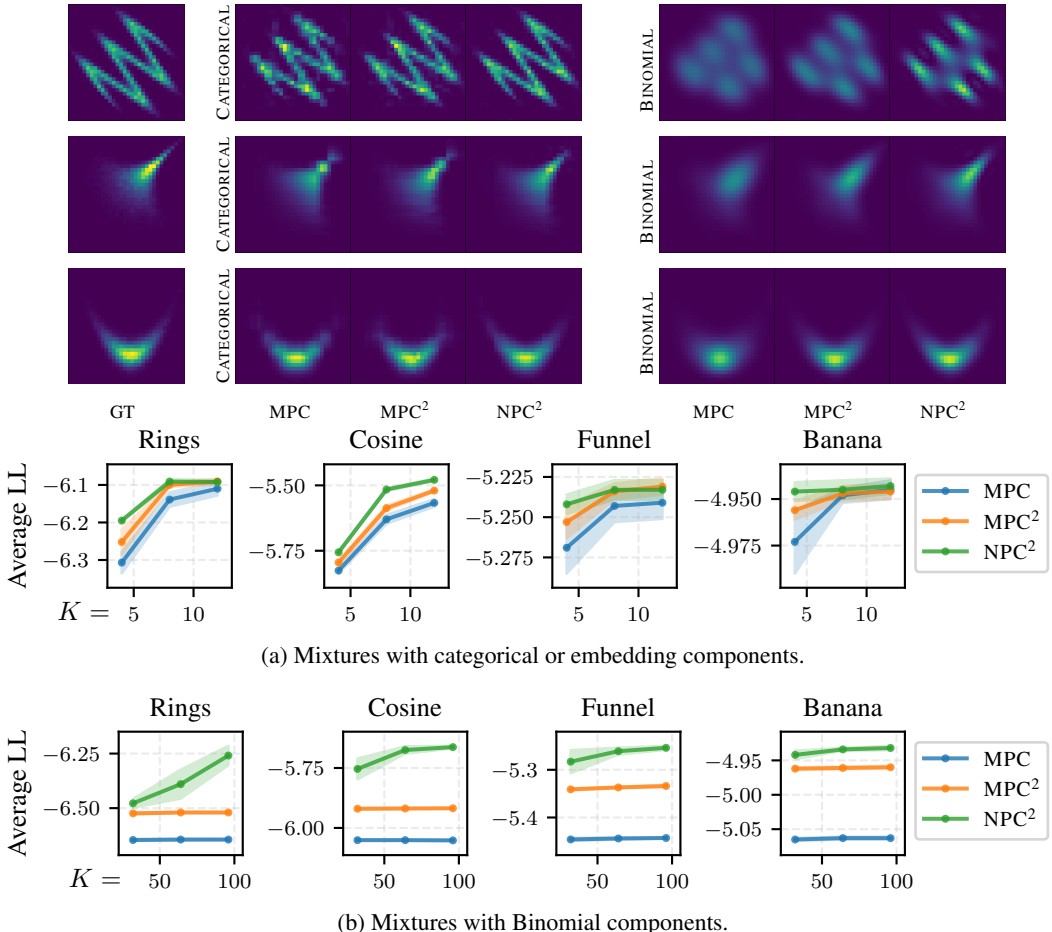

(a) Mixtures with categorical or embedding components.

(b) Mixtures with Binomial components.

Figure H.2: **Negative parameters increases the expressiveness of NPC²s.** From left to right (above) and for each bivariate distribution, we show the ground truth (GT) and its estimation by a monotonic PC (MPC), a squared monotonic PC (MPC²), and a NPC² having input layers computing categoricals (embeddings for NPC²s) and with the same number of parameters. Moreover, we show the average log-likelihoods (and one standard deviation with 10 independent runs) on unseen data achieved by a monotonic MPC, a squared monotonic MPC², and a NPC² with either categorical (a) or Binomial (b) components and by increasing the dimensionality of input layers $K$.

## H.3 UCI CONTINUOUS DATA

**Data sets.** In Sec. 5 we evaluate NPC²s for density estimation on five multivariate UCI data sets (Dua & Graff, 2017): Power (Hebrail & Berard, 2012), Gas (Fonollosa et al., 2015), Hepmass (Baldi et al., 2016), MiniBooNE (Roe et al., 2004) and BSDS300 patches (Martin et al., 2001) by following the pre-processing by Papamakarios et al. (2017). Table H.1 reports their statistics.

|  | | Number of samples | | |
| --- | --- | --- | --- | --- |
|  | $D$ | train | validation | test |
| Power | 6 | 1,659,917 | 184,435 | 204,928 |
| Gas | 8 | 852,174 | 94,685 | 105,206 |
| Hepmass | 21 | 315,123 | 35,013 | 174,987 |
| MiniBooNE | 43 | 29,556 | 3,284 | 3,648 |
| BSDS300 | 63 | 1,000,000 | 50,000 | 250,000 |

Table H.1: **UCI data set statistics.** Dimensionality $D$ and number of samples of each data set split after the preprocessing by Papamakarios et al. (2017).

**Models.** We compare monotonic PCs and NPC²s in tensorized form (Def. 1) for density estimation. The tensorized architecture for both is constructed based on either the *binary tree* (BT) or *linear tree* (LT) RGs (see App. F). In addition, since both RGs are randomly-constructed, we instantiate

eight of them by changing the random seed. By doing so, our monotonic PCs consist of a mixture of tensorized monotonic PCs each defined on a different RG. Conversely, our NPC$^2$s consist of a mixture (with non-negative parameters) of tensorized NPC$^2$s, each constructed by squaring a circuit defined on a different RG. To ensure a fair comparison, monotonic PCs and NPC$^2$s have the exact same structure, but NPC$^2$s allow for negative parameters via the squaring mechanism (see Sec. 3).

**Hyperparameters.** We search for hyperparameters by running a grid search with both monotonic PCs and NPC$^2$s. For each UCI data set, Tables H.2 and H.3 report the possible value of each hyperparameter, depending on the chosen RG. In case of input layers modeling spline functions (see App. E), we use quadratic splines and select 512 uniformly in the domain space.

**Parameters initialization.** We found NPC$^2$s to be more sensible to the choice of the initialization method for parameters than monotonic PCs. The effect of initialization in monotonic PCs is not well explored in the literature, and it is even more unclear for NPC$^2$s as parameters are allowed to be negative. In these experiments, we investigated initializing NPC$^2$s by independently sampling the parameters from a normal distribution. However, we found NPC$^2$s to achieve higher log-likelihoods if they are initialized with non-negative parameters only, i.e., by sampling uniformly between 0 and 1. However, in App. H.5 we show they still learn negative parameters when converging. Note that our work is a first attempt to learn non-monotonic PCs at scale, thus it opens interesting future directions on how to initialize and learn NPC$^2$s, as well as how to regularize them.

**Table H.2: Hyperparameter grid search space for each UCI data set (for BT experiments).** Each data set is associated to lists of hyperparameters: learning rate, the dimensionality of layers in tensorized PCs ($K$), batch size, and whether input layers compute Gaussian likelihoods or spline functions (see App. E).

| Data set | Learning rate | $K$ | Batch size | Input layer |
|---|---|---|---|---|
| Power | | $[32, \ldots, 512]$ | $[512, 1024, 2048]$ | |
| Gas | | $[32, \ldots, 1024]$ | $[512, 1024, 2048, 4096]$ | |
| Hepmass | $[0.01, 0.005]$ | $[32, \ldots, 512]$ | $[512, 1024, 2048]$ | [Gaussian, splines] |
| MiniBooNE | | $[32, \ldots, 512]$ | $[512, 1024, 2048]$ | |
| BSDS300 | | $[32, \ldots, 256]$ | $[512, 1024, 2048]$ | |

**Table H.3: Hyperparameter grid search space for each UCI data set (for LT experiments).** Each data set is associated to lists of hyperparameters: learning rate, the dimensionality of layers in tensorized PCs ($K$), batch size, and whether input layers compute Gaussian likelihoods or spline functions (see App. E).

| Data set | Learning rate | $K$ | Batch size | Input layer |
|---|---|---|---|---|
| Power | | | | |
| Gas | | | | |
| Hepmass | $[0.005, 0.001]$ | $[32, \ldots, 512]$ | $[512, 1024, 2048]$ | [Gaussian, splines] |
| MiniBooNE | | | | |
| BSDS300 | | | | |

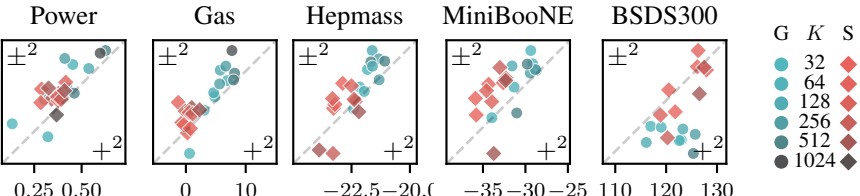

Figure H.3: **Negative parameters make squared non-monotonic PCs more expressive than squared monotonic PCs.** NPC$^2$s ($\pm^2$, vertical) generally achieve higher log-likelihoods than squared monotonic PCs ($+^2$, horizontal) when paired with the same number of units per layer $K$. as shown by the presence of more points in the upper triangle than in the lower triangle for most data sets. Blue circles ● and red diamonds ◆ refer to runs with Gaussian (G) and spline (S) input layers respectively, and darker hues indicate larger $K$. The dashed grey line represents the points of equal log-likelihood for both the NPC$^2$ and the squared monotonic PC.

Table H.4: **Squared non-monotonic PCs can be more expressive than monotonic PCs.** Best average test log-likelihoods and two standard errors achieved by monotonic PCs (MPC) and NPC$^2$s built either from randomized linear tree RGs (LT) or from randomized binary tree RGs (BT) (see App. H.3), when compared to baselines. MPC, MPC$^2$ and NPC$^2$ were experimented with both Gaussian (G) and spline (S) node input layers. † means no values were originally provided.

| | Power | | Gas | | Hepmass | | MiniBooNE | | BSDS300 | |
|---|---|---|---|---|---|---|---|---|---|---|
| MADE | -3.08 ±0.03 | | 3.56 ±0.04 | | -20.98 ±0.02 | | -15.59 ±0.50 | | 148.85 ±0.28 | |
| RealNVP | 0.17 ±0.01 | | 8.33 ±0.14 | | -18.71 ±0.02 | | -13.84 ±0.52 | | 153.28 ±1.78 | |
| MAF | 0.24 ±0.01 | | 10.08 ±0.02 | | -17.73 ±0.02 | | -12.24 ±0.45 | | 154.93 ±0.28 | |
| NSF | 0.66 ±0.01 | | 13.09 ±0.02 | | -14.01 ±0.03 | | -9.22 ±0.48 | | 157.31 ±0.28 | |
| Gaussian | -7.74 ±0.02 | | -3.58 ±0.75 | | -27.93 ±0.02 | | -37.24 ±1.07 | | 96.67 ±0.25 | |
| EiNet-LRS | 0.36 ±† | | 4.79 ±† | | -22.46 ±† | | -34.21 ±† | | † | |
| TTDE | 0.46 ±† | | 8.93 ±† | | -21.34 ±† | | -28.77 ±† | | 143.30 ±† | |
| | G | S | G | S | G | S | G | S | G | S |
| MPC (LT) | 0.51 ±.01 | 0.24 ±.01 | 6.73 ±.03 | -2.05 ±.02 | -22.07 ±.02 | -23.09 ±.02 | -32.48 ±.44 | -37.53 ±.46 | 123.15 ±.28 | 116.90 ±.28 |
| MPC$^2$ (LT) | 0.49 ±.01 | 0.39 ±.01 | 7.06 ±.03 | 0.95 ±.01 | -21.42 ±.02 | -22.24 ±.02 | -29.46 ±.44 | -32.81 ±.47 | — | — |
| NPC$^2$ (LT) | 0.53 ±.01 | 0.43 ±.01 | 9.00 ±.02 | 3.03 ±.02 | -20.66 ±.02 | -21.53 ±.02 | -26.68 ±.42 | -29.36 ±.42 | 112.99 ±.29 | 120.11 ±.29 |
| MPC (BT) | 0.57 ±.01 | 0.32 ±.01 | 5.56 ±.03 | -2.55 ±.02 | -22.45 ±.02 | -24.09 ±.02 | -32.11 ±.43 | -37.56 ±.46 | 121.92 ±.29 | 123.30 ±.29 |
| MPC$^2$ (BT) | 0.57 ±.01 | 0.36 ±.01 | 8.24 ±.03 | 0.32 ±.02 | -21.47 ±.02 | -23.38 ±.02 | -29.46 ±.43 | -33.43 ±.47 | 125.56 ±.29 | 126.85 ±.29 |
| NPC$^2$ (BT) | 0.63 ±.01 | 0.45 ±.01 | 10.98 ±.02 | 3.12 ±.01 | -20.41 ±.02 | -22.25 ±.02 | -26.92 ±.44 | -30.81 ±.54 | 114.47 ±.28 | 128.38 ±.29 |

Table H.5: **Average test log-likelihoods and standard deviation** over five independent runs with random initialization, using the same hyperparameters that brought the results showed in Table H.4.

| | Power | Gas | Hepmass | MiniBooNE | BSDS300 |
|---|---|---|---|---|---|
| MPC (LT) | 0.46 ±0.03 | 7.03 ±0.18 | -22.07 ±0.02 | -31.79 ±0.39 | 126.66 ±5.46 |
| MPC (BT) | 0.53 ±0.03 | 6.16 ±0.56 | -22.42 ±0.45 | -33.30 ±0.98 | 122.77 ±0.71 |
| NPC$^2$ (LT) | 0.42 ±0.11 | 8.97 ±0.08 | -20.67 ±0.05 | -29.58 ±0.29 | 127.58 ±4.66 |
| NPC$^2$ (BT) | 0.62 ±0.01 | 10.55 ±0.39 | -20.48 ±0.11 | -27.64 ±0.44 | 128.45 ±0.52 |

Table H.6: **Best hyperparameters found via grid search**, which were used for achieving results showed in Table H.4. For input layers, G and S respectively denote Gaussian and spline.

| Model | Data set | $K$ | Batch size | Learning rate | Input layer |
|---|---|---|---|---|---|
| MPC (BT) | Power | 512 | 512 | 0.01 | G |
| | Gas | 1024 | 4096 | 0.01 | G |
| | Hepmass | 128 | 512 | 0.01 | G |
| | MiniBooNE | 32 | 512 | 0.01 | G |
| | BSDS300 | 512 | 512 | 0.01 | S |
| MPC (LT) | Power | 512 | 512 | 0.001 | G |
| | Gas | 512 | 1024 | 0.001 | G |
| | Hepmass | 512 | 512 | 0.005 | G |
| | MiniBooNE | 512 | 1024 | 0.005 | G |
| | BSDS300 | 64 | 512 | 0.005 | S |
| NPC$^2$ (BT) | Power | 512 | 512 | 0.01 | G |
| | Gas | 1024 | 512 | 0.01 | G |
| | Hepmass | 256 | 512 | 0.01 | G |
| | MiniBooNE | 32 | 512 | 0.01 | G |
| | BSDS300 | 128 | 512 | 0.01 | S |
| NPC$^2$ (LT) | Power | 512 | 512 | 0.001 | G |
| | Gas | 512 | 512 | 0.001 | G |
| | Hepmass | 256 | 512 | 0.001 | G |
| | MiniBooNE | 128 | 2048 | 0.005 | G |
| | BSDS300 | 32 | 1024 | 0.001 | S |

## H.4   LARGE LANGUAGE MODEL DISTILLATION

**Data set.** Given $p^*(\mathbf{x})$ the distribution modeled by GPT2 over sentences $\mathbf{x} = [x_1, \dots, x_D]$ having maximum length $D$, we aim to minimize the Kullback-Leibler divergence $\mathrm{KL}[p^* \mid p]$, where $p$ is modeled by a PC. Minimizing such divergence is equivalent to learn the PC by maximum-likelihood on data sampled by GPT2. Therefore, following the experimental setting by Zhang et al. (2023) we sample a data set of 8M sentences using GPT2 having bounded length $D = 32$, i.e., with a

maximum of $D = 32$ tokens. Then, we split such sentences into training, validation and test set having proportions 0.85/0.05/0.10, respectively.

**Models.** Then, we learn a monotonic PC and a NPC$^2$ as tensorized circuits whose architecture is determined by a linear tree RG (Def. 2), i.e., a region graph that recursively partitions each set of finitely-discrete variables $\{X_i, \dots, X_D\}$ into $\{X_i\}$ and $\{X_{i+1}, \dots, X_D\}$ for $1 \leq i \leq D - 1$ (e.g., see Fig. 2a). This is because we are interested in exploiting the sequential dependencies between words in a sentence. By enforcing monotonicity, we recover that the monotonic PC is equivalent to an inhomogenous hidden Markov model (HMM), and that that NPC$^2$ corresponds to a Born machine (see App. B.4.1 for details).

**Hyperparameters.** All PCs are learned by batched stochastic gradient descent using Adam (Kingma & Ba, 2015) as optimizer with batch size 4096, and we continue optimizing until either the validation loss does not improve after three consecutive epochs or the maximum budget of 200 epochs has been reached. We perform multiple runs by exploring combinations of learning rates and initialization. For monotonic PCs, we run experiments by choosing learning rates in $\{5 \cdot 10^{-3}, 10^{-2}, 5 \cdot 10^{-2}\}$ and initializing parameters by sampling uniformly in $(0, 1)$, by sampling from a standard log-normal distribution, and from a Dirichlet distribution with concentration values set to 1. Similarly for NPC$^2$s, we run experiments by choosing the same learning rates for monotonic PCs, but using different initialization methods. Since squaring results in much larger outputs when compared to monotonic PCs, we initialize NPC$^2$s such that the magnitude of parameters is relatively small. That is, in addition to sampling uniformly in $(0, 10^{-1})$, we also initialize the parameters by sampling from a normal distribution having mean 0 and standard deviation $10^{-1}$. By doing so, we initialize an approximately even number of positive and negative parameters. Moreover, we also experiment by initializing parameters by sampling from a normal distribution with mean $10^{-1}$ and standard deviation $10^{-1}$, which initializes more parameters to be positive.

**Results.** For increasing layer dimensionality, we group runs having different learning rate and initialization method together and show the achieved log-likelihoods in Fig. 5. Moreover, for layer dimensionalities $K \leq 256$, we report the double of log-likelihood points by repeating the runs with a different seed. Then, we perform statistical tests to assess the significance of NPC$^2$s achieving higher log-likelihoods than monotonic PCs on the test data, and show the p-values in Table H.7.

| $K =$ | 32 | 64 | 128 | 256 | 512 | 1024 |
|---|---|---|---|---|---|---|
| p-value $=$ | 1.0000 | 0.1071 | **0.0019** | **0.0020** | **$< 0.0001$** | **$< 0.0001$** |

Table H.7: **Statistical significance of NPC$^2$s achieving higher likelihoods on LLM distillation.** We perform a one-sided Mann-Whitney U test between the log-likelihoods achieved by NPC$^2$s and monotonic PCs on the test data (see also Fig. 5), using a total of 36 runs for layer dimensionalities $K \leq 256$ and 18 runs for $K > 256$. We highlight the p-values that are consistent with a 99% confidence interval in bold.

## H.5   HISTOGRAMS OF LEARNED PARAMETERS

In Fig. H.4, we show the parameters of both monotonic PCs and NPC$^2$s learned in our experiments (Sec. 5), i.e., distribution estimation on UCI data sets (Table H.1) and sentences sampled from GPT2. Even though we initialize the parameters of NPC$^2$s to be non-negative in our experiments (i.e., by sampling from a uniform distribution located on the non-negative side), they still end up learning negative parameters. In particular, Fig. H.4 shows the histograms of sum layers parameters of monotonic PC and NPC$^2$s learned on UCI data sets having the same model size, i.e., with layer dimensionality $K = 512$ for Power and Gas, $K = 128$ for Hepmass and MiniBooNE, and $K = 64$ for BSDS300. For the rest hyperparameters, we choose batch size 512, learning rate $10^{-2}$, quadratic splines as input layers, and build a mixture of tree-shaped circuits (as described in App. H.3). For the models learned on GPT2 sentences, we use learning rate $10^{-2}$ and uniform initialization with non-negative values (see also App. H.4). Note that for NPC$^2$s we report the parameters of the circuit *after* being squared with Alg. 1, thus resulting in a quadratic increase in the number of parameters.

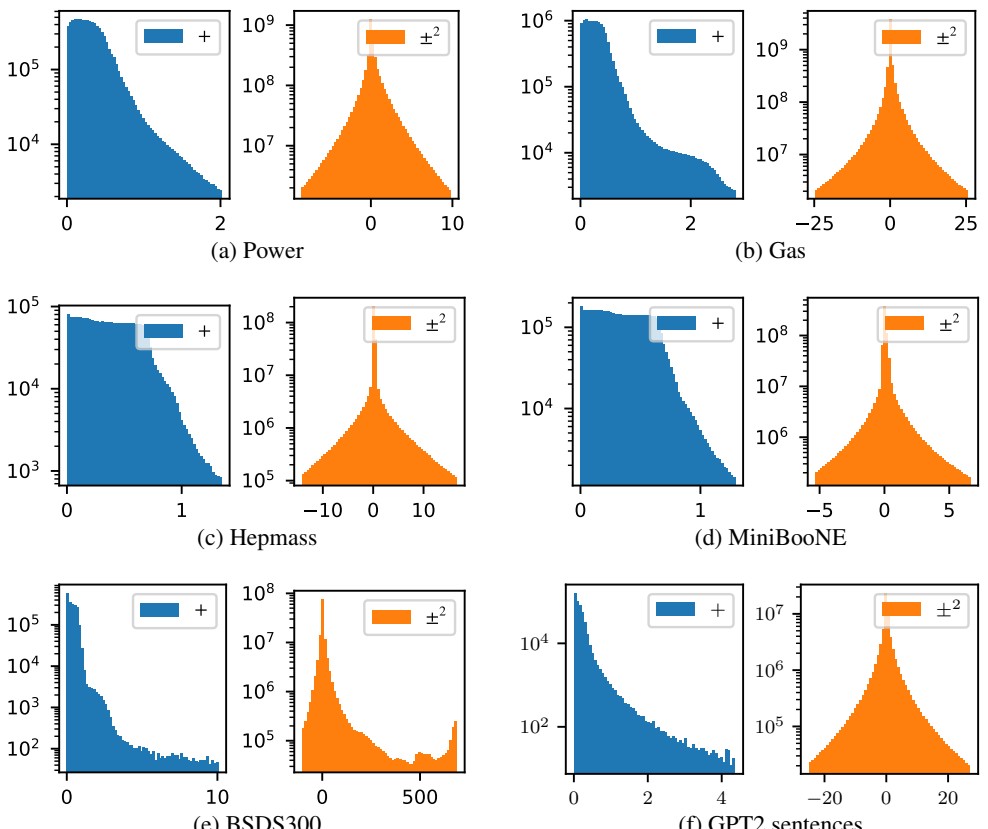

Figure H.4: **NPC²s learn negative parameters with non-negative initializations.** Histograms containing the 99% interquartile range of the sum layers parameters of monotonic PCs ($+$, blue) and NPC²s *after* being squared ($\pm^2$, orange) that are learned on UCI data sets (a-e) (App. H.3) and on sentences sampled from GPT2 (f) (App. H.4). Even if the chosen NPC²s are initialized with non-negative parameters (see App. H.5), they converge to a model instance with negative parameters.

