# OpenReview forum: "Subtractive Mixture Models via Squaring: Representation and Learning"
_ICLR.cc/2024/Conference — ICLR 2024 spotlight_

### Official Review · Reviewer_Bp5w · 2023-10-27

**Soundness:** 4 excellent
**Presentation:** 4 excellent
**Contribution:** 3 good
**Rating:** 8
**Confidence:** 4

**Summary:**

The paper proposes a way to learn and represent mixture models with negative weights by squaring the linear combination defined by these models. The resulting model is cast into the Probabilistic Circuits (PCs) framework and thus further extended to deep mixture models, resulting in the main contribution of the paper, squared non-monotonic PCs (or NPC$^2$s). NPC$^2$s are theoretically proven to be more expressive efficient than regular (or monotonic) PCs, which translates to strong empirical results in a number of benchmarks.

**Strengths:**

The work is novel and certainly relevant for the Tractable Probabilistic Models community, since it adds a new and provenly more expressive model to the class of Probabilistic Circuits. More importantly, the main ideas are well developed in the text and thoroughly analyzed both theoretically and empirically, allowing for a well-rounded understanding of this new class of models. The text itself is well written and easy to follow, and the related work develops important connections to other methods and models in the literature of Probabilistic Circuits and beyond.

**Weaknesses:**

The paper is very well executed, and honestly I cannot think of any major flaws or possible improvements besides the couple of questions I outline below.

**Questions:**

1. Unless I missed it, there is no mention of how the number of parameters of the MPCs used in the experiments compare to that of MPC$^2$ and NPC$^2$. I assume the authors used the same number of parameters for all models, which is probably the most natural comparison, but it would be interesting to compare the squared models with an MPC of size equal to that of the unrolled model (e.g. as in the right hand side of Figure 1). I imagine MPC$^2$ and NPC$^2$ would still perform better because, despite being less flexible than the unrolled MPC, they effectively share weights among components, which probably facilitates learning. Would the authors share the same opinion? Weight sharing might be important in practice, and the paper could benefit from exploring this aspect a bit further in the text or experiments.
2. How does the optimization of NPC$^2$ with SGD compare to that of regular MPCs? Some would argue that optimization is one of the main bottlenecks holding PCs back, since they tend to converge relatively fast, probably quickly getting stuck in local minima. Have the authors observed any notable differences in convergence or stability of the optimization process of NPC$^2$s as compared to MPCs? Even though NPC$^2$s are provenly more expressive than regular MPCs, their utility is heavily tied to how easy they are to learn, and it would be useful to have more insights in that sense.

Small remarks:
- I believe the signed logsumexp trick, or a very similar solution, was already used in previous works to compute expectations of arbitrary functions using PCs [1, 2]. The PCs used there did not have negative weights, of course, but the signed logsumexp trick was needed to propagate negative values through the networks since the functions one would compute the expectation of could take negative values. I am not sure the trick is actually mentioned in the papers, but it is certainly used in their respective implementations.
- Line 227: “they” is repeated.
- Line 303: The “to” after “answer” is not necessary.

---

> ### Author Response · Authors · 2023-11-19
>
> We thank the reviewer for appreciating the novelty of our work and the theoretical connections with other tractable probabilistic models. We are also keen on citing the additional work (but references seem missing!)
>
> >There is no mention of how the number of parameters of the MPCs used in the experiments compare to that of MPC^2s and NPC.
>
> In the experiments we always compare MPCs and NPC^2s having the same number of learnable parameters. We added a sentence in the latest version of the paper in order to make this explicit. We also compare NPC^2 against squared MPCs (MPC^2s) in order to disentangle the effects given by squaring and the negative parameters.
>
> >I imagine MPC and NPC^2 would still perform better because [...] they effectively share weights among components, which probably facilitates learning. Would the authors share the same opinion?
>
> We perform experiments with squared MPCs (MPC^2s) as well, therefore taking into account parameter sharing for monotonic circuits. Table H.4 (in the latest revision) brings evidence that the parameters sharing given by squaring is beneficial for density estimation even in monotonic circuits (e.g., see results of MPC^2s against MPCs on Gas, Hepmass, Miniboone). And it also supports our claim around negative parameters being the key factor in boosting performance, as NPC^2s outperform MPC^2s.  Theoretically investigating if parameter sharing creates an expressive efficiency advantage or how its influence on learning is interesting future work.
>
> >How does the optimization of NPC with SGD compare to that of regular MPCs? Have the authors observed any notable differences in convergence or stability of the optimization process of NPCs as compared to MPCs?
>
> The fast convergence to some local minimum with MPCs that the reviewer mentioned has been observed on UCI data sets. By contrast, NPC^2s generally require more epochs to converge than MPCs, but they achieve higher log-likelihoods on both training and test data.
>
> Regarding the stability of optimization, we found NPC^2s to be more sensible to the parameters initialization method than MPCs. In general, choosing initialization is already tricky for MPCs, and it is even more difficult for NPC^2 as parameters are allowed to be negative. In our experiments we investigated initializing NPC^2s by sampling the parameters from a normal distribution independently. However, we found NPC^2s to achieve higher log-likelihoods if they are initialized with non-negative parameters only (i.e., by sampling uniformly between 0 and 1). Our paper is a first attempt to learn non-monotonic circuits in a principled way and at scale.  We believe it can open up plenty of interesting future directions on how to initialize, regularize and learn NPC^2s.
>
> >I believe the signed logsumexp trick [...] was already used in previous works to compute expectations of arbitrary functions using PCs [1, 2].
>
> Can you please provide the references [1, 2] ? We will definitely cite them in our work.
>
> >Typos
>
> Thank you! We fixed the typos in the latest version of the paper.

---

> > ### Comment · Reviewer_Bp5w · 2023-11-21
> >
> > I thank the authors for the clarifications. I am satisfied with the answers to my questions.
> >
> > > Regarding the stability of optimization, we found NPC^2s to be more sensible to the parameters initialization method than MPCs. In general, choosing initialization is already tricky for MPCs, and it is even more difficult for NPC^2 as parameters are allowed to be negative. In our experiments we investigated initializing NPC^2s by sampling the parameters from a normal distribution independently. However, we found NPC^2s to achieve higher log-likelihoods if they are initialized with non-negative parameters only (i.e., by sampling uniformly between 0 and 1).
> >
> > That is an interesting practical insight that might be highly relevant for future work on NPCs. There is a brief discussion on parameter initialization at the end of the appendix, but I think the authors should include the insights mentioned here in the paper as well, namely that NPCs are more sensitive to the initialization and benefit from non-negative initial parameters.
> >
> > > Can you please provide the references [1, 2] ? We will definitely cite them in our work.
> >
> > I am sorry for the missing references in my original review. I included them below.
> >
> > [1] Mauá, Denis Deratani, et al. "Robustifying sum-product networks." International Journal of Approximate Reasoning 101 (2018): 163-180.
> >
> > [2] Correia, Alvaro HC, and Cassio P. de Campos. "Towards scalable and robust sum-product networks." Scalable Uncertainty Management: 13th International Conference, SUM 2019, Compiègne, France, December 16–18, 2019, Proceedings 13. Springer International Publishing, 2019.

---

> > > ### Author Response · Authors · 2023-11-22
> > >
> > > >There is a brief discussion on parameter initialization at the end of the appendix, but I think the authors should include the insights mentioned here in the paper as well, namely that NPCs are more sensitive to the initialization and benefit from non-negative initial parameters.
> > >
> > > We agree with the reviewer and updated the paper by including our discussion on parameters initialization for NPC^2s in Appendix H.3.
> > >
> > > >I am sorry for the missing references in my original review. I included them below.
> > >
> > > Thank you! We cited those works in the same paragraph.

---

### Official Review · Reviewer_Lqff · 2023-10-27

**Soundness:** 3 good
**Presentation:** 2 fair
**Contribution:** 3 good
**Rating:** 8
**Confidence:** 2

**Summary:**

Mixture models (herein called MPC) are probability density functions over a domain $X$ that are composed by adding together multiple simpler probability density functions. The authors propose a more general framework, (called NPC), where any functions over the domain $X$ (or specific dimensions of $X$) may be multiplied, added, subtracted together and finally squared to return a non-negative scalar which, under the right conditions the authors prove, can also be normalised yielding a probability density. Given the same number of parameters, the proposed NPC models have far more expressive capability than traditional MPCs, in fact proven exponentially more capacity.

The authors describe a range of concepts, rooted graphs, tensor circuits, to lead to the conditions for a given NPC to be marginalizable. Theoretical relationships with related work is described and finally experiments demonstrate the efficacy of the propoosed approach.

**Strengths:**

- extensive theoretical evaluation
  - proof of conditions for normalization which significantly affects model design
  - proof of exponential expresivity
  - proofs of connections to related methods

- simplicity
  - take some functions over the space $X$, functions over disjoint subsets of dimensions
  - multiply them (across disjoint groups)
  - add them together (functions within a disjoint group)
  - sum the output and square it to get a single non-negative scalar value
  - if we followed the rules, we can find the partition function/normalising constant in one forward pass of the function
  - I feel this is very intuitive and surprisingly simple, and most importantly  avoids difficult normalisation (e.g. MCMC) while yielding a big improvement in expressiveness over standard mixture models. Analytic normalisation also enables conditioning. Sampling is briefly mentioned in the Appendix and appears to be one point to be harder than MPCs.

**Weaknesses:**

While I feel confident I understood the method paper and the paper, I am not familiar with the surrounding literature hence my comments are mainly on the general practicalities and the paper presentation.

## Major Comments

The main four big questions I have mostly relate to practical considerations
- How do I normalise $c(X)$? Much of section 3 is building foundations to finally reach proposition 1 which successfully solves the issue of normalization (which is a significant strength of the paper in my view)
- For a given dataset, how do I find the rooted graph? The issue is not so carefully discussed, and presumably there is no theoretical or provable result and in practice one must simply employ some sort of architecture search. If not already, this should be clarified in the main paper as I feel it may be an obstacle for practitioners.
- Given the exponentially more expressiveness, could this easily lead to overfitting? A quick word search in the document doesn't yield results, this is not mentioned once?
- How do I generate samples? This is described in the appendix and appears to be not as easy compared to MPCs,

I understand if the authors would like to argue this is a more theoretical paper and such practicalities are beyond scope.

## Minor Comments (Presentation)

### Content Density
Upon first reading, I was exhausted by section 5, however upon second reading the paper made much more sense.
  - Sections 1, 2 were simple and easy to follow
  - section 3 was hard work on first reading but very clear on second reading (see details below)
  - section 4 contains multiple short sharp deep dives into a range of related fields.
  - section 5 was very short and I personally didn't truly understand the benchmarks nor get a feel for implementing the method or its practicalities or failure modes, (e.g. overfitting, sensitivity to rooted graph, sampling)

Given the main paper introduces a range of concepts, then proposes a new method, provides proven results and proves connections to related fields and then benchmarks, I feel like this is a (very nice) journal paper that was heavily compressed into 9 pages and all of the overflow was placed in the appendix.

The theoretical treatment is extensive. The authors may consider moving some of the less significant content regarding other works in Section 4 to the appendix in order to extend section 5 with more "hands on" details about the using NPCs, e.g. a worked illustrative example or failure modes or sensitivity to the rooted graph.

### Section 3 detailed comments
I understand computational graphs have nodes that are operations and the links are tensors,
- Definition 1 was very hard to follow, the ambiguous notation that $\ell$ represent a layer as well as a numerical output tensor from a layer.
- Figure 2 b, c, blur the boundary of "nodes" and "edges", there are rectangles with operations, and there are volumes between them also with operations (hadamard product and $W$). Even now I struggle to parse these diagrams.

**Questions:**

- In Figure 2,b,. the sum layer must output dimension $S=K$ in order to be accepted by the following product layer?
- how does one find the rooted graph/division partition or variables? Try a handful of graphs and choose the maximum likelihood graph?
- there is much discussion on tensored circuits which have a tensor output, I presume the final layer is a sum to scalar layer?
- the authors justify the naming of "tensorized circuits" as a way to encompass and simplify other methods. If I understand correctly, these  are standard computational graphs, the bread and butter of all pytorch or tensorflow users, is a new name really required?

---

> ### Author Response · Authors · 2023-11-14
>
> We thank the reviewer for appreciating the theoretical contributions of our work, as well as its simplicity. Our paper is not merely theoretical, but its theoretical contribution is definitely a major strength. Indeed, we present one of few lowerbounds for non-monotonic circuits, as well as important connections to PSD models and tensor networks. Bridging these previously disconnected literatures can greatly propel future research in PCs (and viceversa). Concerning presentation, we agree with the reviewer and we moved Section 4.2 to Appendix G, and better describe how we build region graphs in the main text and in Appendix F and Figure F.1. We answer the posed questions below, and we are happy to further clarify any additional point.
>
> >How do I normalise c(X)?
>
> We assume you are referring to normalising $c^2(X)$. Computing its partition function Z is done by recursively decomposing the integral and by exploiting the structure of $c^2$. E.g., in Eq. 3 Z can be computed by i) computing integrals over products of components, and then ii) a linear combination thereof. This method applies to deep squared NPCs, where the integral is recursively decomposed until it is first applied on products of functions at the input layers (for example see input layer labelled with $f_i(X_3)f_j(X_3)$ in Figure 2c). Then, these computations are propagated to the inner layers, which are evaluated only once. Finally, the scalar output will be exactly Z of $c^2(X)$.
>
> >[...] could this easily lead to overfitting?
>
> An increased expressiveness does not imply that the model will overfit more easily. As many other machine learning models overfitting is possible, but the risk can be reduced by limiting the number of parameters.
> >How do I generate samples?
>
> Sampling can be done in an autoregressive fashion. We choose an ordering of variables and sample one at a time in that order. To sample the i-th variable $X_i$, we fix all the previous i - 1 variables that have been already sampled and sample from the conditional $p(X_i \mid X_1,\ldots X_{i-1})$. This requires evaluating the tensorized circuit as many times as the number of variables and not only once as for MPCs. However, since we are actually evaluating the same tensorized circuit, many inner computations can be cached and reused. This speeds-up sampling, but requires engineering effort that is out of the scope of this work.
>
> >[...] what is a worked illustrative example or failure modes?
>
> In Section 5 we already discuss some failure scenarios of NPC^2s. E.g., we show that there is little advantage in having negative parameters in NPC^2s if the input components are already flexible enough such as Categoricals. We illustrate this aspect in Figure 3. Also, we did not find any significant improvement with respect to monotonic PCs on image data, as we mention in L326-328.
>
> >[...] the sum layer must output dimension S=K?
>
> Sum layers can output a vector (of size S) that is smaller or bigger than their input (of size K). Since the inputs of element-wise product layers must be of the same dimensionality, it means that when a sum layer feeds a product layer (like in Fig 2b) then its output has the same size as the product layer. However, for example, in Fig 2b we can have input layers that are smaller than the sum layers, hence some sum layers will necessarily have S > K.
>
> >How to find the region graph?
>
> There are various ways coming from the PCs literature. For example, one can recursively split sets of variables randomly until no further splitting is possible [A]. Alternatively, it can be learned from data by performing independence tests to split variables [B]. In case of image data one can split patches of pixels [C].
>
> >Is the final layer a sum to scalar layer?
>
> Yes, the output is a scalar. However, one can also have tensorized circuits with a tensor output. In fact, our Algorithm 1 operates on such tensorized circuits after the first recursive step.
>
> >[...], is a new name really required?
>
> Circuits are computational graphs, with special structural constraints. Therefore sometimes they are referred to structured neural networks [D]. We call our circuits “tensorized” as we define them in a layer-wise fashion via tensor operations, as opposed to the classical way to define circuits that is unit-wise [E]. For example, Figure A.1 shows how scalar units in circuits can be tensorized to form layers. Modern circuit architectures are indeed tensorized [A] [F]
>
> [A] Peharz, Random Sum-Product Networks: A Simple and Effective Approach to Probabilistic Deep Learning, 2019
>
> [B] Gens Learning the Structure of Sum-Product Networks, 2013
>
> [C] Poon, Sum-product networks: A new deep architecture, 2011
>
> [D] Vergari, Tractable probabilistic models: Representations, algorithms, learning, and applications, 2019
>
> [E] Choi, Probabilistic Circuits: A Unifying Framework for Tractable Probabilistic Models, 2020
>
> [F] Peharz, Einsum Networks: Fast and Scalable Learning of Tractable Probabilistic Circuits, 2020

---

> > ### Author Response · Authors · 2023-11-20
> >
> > We hope all your concerns have been addressed by our answers above. We are very keen to discuss any other aspects and to solve additional issues in order to obtain a full acceptance.

---

### Official Review · Reviewer_5G72 · 2023-11-02

**Soundness:** 3 good
**Presentation:** 2 fair
**Contribution:** 2 fair
**Rating:** 6
**Confidence:** 3

**Summary:**

Mixture models traditionally are represented as convex combinations of simpler probability distributions. This paper proposes loosening this constraint to any linear combination, and squaring the result at the end to ensure non-negativity. This modification is then applied to probabilistic circuits (by allowing negative weights in sums, and then squaring). Theoretical and empirical analysis confirm:

1. Better distribution approximations for a given number of parameters (with a theoretical example showing exponential separation)
2. Preservation of smoothness and decomposability when converting a PC to a squared NPC.

**Strengths:**

- Simple and effective idea
- Empirical results show better performance than baseline on some tasks

**Weaknesses:**

- Paper's motivation can be stronger. e.g. add a real world motivating example. It would be interesting to see how the better density estimation can be used for an improved downstream task as well.
- The NPCs use fewer parameters but in a more complex way. What is the impact of this on training cost. This question is not explored empirically.

**Questions:**

Questions/Suggestions:
- GPT2 distillation experiment should compare other tractable models
- Exponential separation is established theoretically for a restricted class of functions (unique disjointness). Is it possible to establish that for a more general class of distributions?
- Definition A.1 in Appendix A seems to have a typo. The sum nodes should use sc(n_i), instead of sc(c_i)?

---

> ### Author Response · Authors · 2023-11-16
>
> We thank the reviewer for appreciating the simplicity and effectiveness of our idea and the empirical evidence that shows an improved distribution estimation task. We believe our answers below can address all the raised concerns. We are happy to clarify further points and to run further experiments if the reviewer points us to specific cases.
>
> >It would be interesting to see how the better density estimation can be used for an improved downstream task as well.
>
> Our experiment on distilling GPT2 already shows the potential of NPC^2 for a downstream task. That is, [A] proposes how models such as PCs (and hence also NPC^2) can be used in combination with large language models (LLMs) for the downstream task of constrained text generation. In particular, Figure 3 in [A] shows that the log-likelihood highly correlates with the text generation quality (measured as BLEU-4 scores). Therefore, higher log-likelihoods would directly translate to better performances on the mentioned downstream task.
>
> >[...] What is the impact of this on training cost?
>
> The increased complexity of NPC^2 during training arises only from the computation of the partition function Z, which however can be greatly amortised over large batches as we show in Appendix C and Figure C.1. There, one can see that for NPC^2 the cost of computing Z is similar to the cost of evaluating the circuit once on a batch of data. This is because the complexity of Z does not depend on the batch size and therefore its cost can be greatly amortised.
>
> In the rebuttal revision we added Figures C.2 and C.3, which compare the actual computational cost of training the monotonic PCs and NPC^2 on UCI data sets. In these figures, we show that NPC^2 adds little overhead during training in most configurations, as computing the partition function Z is comparable to evaluating $c(x)$ or $c^2(x)$ on a batch of data. Concerning the likelihoods gains the overhead for training on Gas results in a NPC^2 achieving a x2 improvement on the final test log-likelihood. Moreover, on high-dimensional data such as BSDS300, NPC^2s are actually faster as they require fewer parameters and achieve higher log-likelihoods. We also report an analysis for the largest possible models (worst-case) in Figure C.3, and confirm that the cost of computing Z is still comparable to evaluating $c(x)$.
>
> >GPT2 distillation experiment should compare other tractable models
>
> If the reviewer can specify other tractable models or an available implementation of flows for text data, we will gladly run the additional experiments. Note that the normalising flows we have in the table of Figure 4 operate on continuous variables and therefore cannot be applied on discrete data such as text in our GPT2 distillation experiment.
>
> >Exponential separation is established theoretically for a restricted class of functions. Is it possible to establish that for a more general class of distributions?
>
> Exponential separations as in our result are often very tricky. First, it is often hard to find candidates for separations: many functions turn out to be too easy or too hard, in the sense that they are either easy to compute in both models one wants to separate or hard in both. And even when a candidate function for being a separating example is found, it also has to be amenable to theoretical analysis. This often requires very regular, structured characteristics of the function that allow a good understanding.
> Furthermore, note that we never have access to the real distribution that has generated the data. And even if we had access to it, it would require the above regularity conditions to make a theoretical claim. The combination of these effects often makes the functions for which one can actually show separations very specific!
>
> We remark that in [D] there are quite a number of open questions about exponential separations between non-monotonic circuits. We hope our new lower bound can help build other results that close these open questions. So while we would of course be delighted to show separation results for more classes of functions, we are very happy with the result we can show.
>
> >Definition A.1 in Appendix A seems to have a typo
>
> Yes, thank you! We fixed it in the updated revision.
>
> [A] Zhang et al.,Tractable Control for Autoregressive Language Generation, 2023
>
> [B] Hoogeboom et al., Integer Discrete Flows and Lossless Compression, 2019
>
> [C] Tran et al., Discrete flows: Invertible generative models of discrete data. 2019
>
> [D] de Colnet, Mengel, A Compilation of Succinctness Results for Arithmetic Circuits, 2021

---

> ### Author Response · Authors · 2023-11-20
>
> We hope all your concerns have been addressed by our answers and the new figures we added in the latest paper version. We are willing to run more experiments if the reviewer specifies which model to evaluate. In any case, we remain open to solve any other pending issue to achieve a full acceptance.

---

### Official Review · Reviewer_uVYn · 2023-11-04

**Soundness:** 4 excellent
**Presentation:** 3 good
**Contribution:** 4 excellent
**Rating:** 8
**Confidence:** 3

**Summary:**

Motivated by mixture models, the paper investigates a class of functions that is a squared mixture of arbitrary functions with potentially negative weights and functions that do not necessarily represent a density function. After motivating this in the "shallow" regime, the authors propose extensions to deep mixtures based on tensorized circuits.

**Strengths:**

- [S1]: Originality -- as far as I can judge it's a novel and very interesting
- [S2]: Significance -- seems to often work better than mixture models and other alternatives such as flows
- [S3]: Clarity -- While the part related to tensor computations is a bit dense and could benefit from a more higher-level treatise, the paper is clearly written

**Weaknesses:**

- [W1]: Missing discussion / limitations: Maybe I overlooked this, but I could not find an actual discussion about the restriction of the approach, e.g.,
    + what are the limitations of the approach?
    + how restrictive is the induced functional form by using squared functions?
    + how expressive is the approach in the shallow or small-K setting? (Fig. 5 e.g. indicates that $\pm$ is worse for small $K$)
    + is it possible to extend the approach to a conditional setup?
- [W2]: Experiments: The paper addresses the computational costs of the approach from a theoretical point of view and even provides some empirical evidence for the computation of the normalization constant, but empirically investigating a couple of scaling aspects such as
    + the scaling in $D$ or
    + a comparison with other appraoches such as the MAFs in terms of runtime

    would provide further valuable insights (e.g. related to "fairness" when tuning different methods).
- [W3]: Presentation (minor): Some of the graphics are rather hard to read:
    + relatively small and dense (Fig. 4)
    + the x or y-axis labels are sometimes missing (Fig. 4) or hard to find (not centered; Fig. 5, C1)

**Questions:**

- See [W1]
- Do authors have additional insights on [W2]?

---

> ### Author Response · Authors · 2023-11-17
>
> We thank the reviewer for pointing out the novelty of our approach and the significance of our empirical results. We answer below to the comments raised.
>
> >What are the limitations of the approach?
>
> We already show two limitations about NPC^2s in our paper. First, we show that negative parameters are not really useful if we use flexible yet expensive components, such as Categoricals (see L309-320 and log-likelihoods shown in F.2 (a) (in the rebuttal revision they are now shown in H.2)). Second, we highlight the aspect that NPC^2 cannot support tractable MAP inference without losing the benefits given by negative parameters (see discussion in Section 4.2 and Proposition 4).
>
> Finally, to complement our time and memory benchmarks shown in Figure C.1, we added Figures C.2 and C.3 in the latest paper version. We show that training NPC^2s might require slightly more time and memory in some cases (see also our answers to reviewers 5G72 and nUGh).
>
> >How restrictive is the induced functional form by using squared functions?
>
> We already know that Gaussian mixture models (GMMs) and monotonic PCs are universal approximators. Moreover, we showed that NPC^2s can require fewer parameters than monotonic PCs (Theorem 1), i.e. they are more expressive efficient. Showing whether other models can be more expressive efficient than NPC^2 is an open problem. Nevertheless, our empirical analysis suggests squaring is not restrictive for density estimation on real world data sets (see Figure 4).
>
> >How expressive is the approach in the shallow or small-K setting?  (Fig. 5 e.g. indicates that +- is worse for small K)
>
> We experiment with NPC^2s on 2D artificial data sets with small K values, which results in shallow models that are more expressive than the monotonic PCs (MPCs) that are equivalent in size.
>
> In Figure 5, the worse results on the test data for very small $K$ (such as $K=32$) indeed requires further investigation. We believe it is because smaller models are much more sensitive to initialization, and we believe initialization is an important aspect in NPC^2 that definitely deserves future work. See also our answers to Reviewers nUGh (about our results showed in Figure 5) and Bp5w (about difficulties in training NPC^2s).
>
> >Is it possible to extend the approach to a conditional setup?
>
> Yes, e.g., one can construct a NPC^2 modelling a conditional distribution $p(Y\mid X)$ for some labels $Y$. To do so, you can learn a neural network that takes the features $X$ as inputs and outputs the real-valued parameters of a NPC^2 modelling a probability distribution over $Y$. This idea has already been investigated for monotonic PCs, see [A].
>
> >[...] empirically investigating a couple of scaling aspects such as the scaling in D [...]
>
> In the rebuttal revision we added Figures C.2 and C.3, showing how training monotonic PCs and NPC^2s scale when increasing the data set dimensionality. The results show that squaring in NPC^2s adds little overhead in terms of time and memory in most configurations.
> See also our responses to Reviewers 5G72 and nUGh.
>
> >Presentation (minor): Some of the graphics are rather hard to read [...]
>
> We want to clarify that the scatter plots in Figure 4 lack labels on the x-axis because they would be exactly the one of the y-axis. In fact, each grey diagonal separates the graph into two even trapezoids. Furthermore, the y-axes of the two plots in Figure C.1 are shared.
> In the latest revision, we updated the captions of these Figures to make this explicit.
>
> [A] Shao, Conditional Sum-Product Networks: Imposing Structure on Deep Probabilistic Architectures, 2019

---

### Official Review · Reviewer_nUGh · 2023-11-04

**Soundness:** 3 good
**Presentation:** 2 fair
**Contribution:** 3 good
**Rating:** 6
**Confidence:** 2

**Summary:**

The authors consider the problem of learning a mixture model where the separate components do not have to be positive.

This can be naively done by squaring the additive MM but this is computationally inefficient.

The authors develop a method based on probabilistic circuits allowing to square different model structures without excessive computational cost.

**Strengths:**

The paper is reasonably clear and proposes simple yet interesting idea which appears to work well on selected synthetic/small scale experiments.

The authors provide the code for the experiments (which I did not reviewed).

The figure in page 1 nicely summarises the benefit of relaxing the requirement of positive components. Overall the figures in the paper help to understand the introduced concepts.

I think the paper is an interesting read.

**Weaknesses:**

The clarity of the paper in pages 4,5,6 could be improved, the presentation is very dense and discusses multiple threads. The paper would benefit from focusing on core ideas and describing them in more detail while the less important parts could be moved to the appendix.

I have concerns that a few points in the paper are overselling the method (i.e the result in Fig 5. on test data appears very small if statistically significant at all but using ^2 introduces additional computational cost). I would welcome the balanced discussion describing advantages and disadvantages of the method.

The authors do not discuss in detail how much additional computational cost is needed to achieve these results (a plot log-likelihood improvement vs CPU time would make the paper stronger).

Error bars in Figure 2 would help to understand the significance of empirical results.

In my eyes, the empirical improvements warranted by the proposed method are rather small and mostly shown on synthetic data.

The authors somewhat addressed three different questions in the empirical section but I feel it would be nicer to provide strong evidence for just one question: Does NPC^2 provide strong gains in performance without substantial increase in computational cost?

In Figure 4 the authors show that while NPC^2 outperforms MPC for LT and BT separately, for the cross comparison NPC^2(LT) vs MPC(BT) the latter can be better (similarly in table F5). This begs the question: is the RG doing the heavy lifting? If so, more empirical analysis would be helpful.

The authors should also elaborate on the selection of RG for improved clarity.

Why the differences reported in F2(a) are so small?

Can the authors elaborate on the statement 105-106 regarding batching? I appears not fully clear to me. I cannot see how one can perform batching in (4) without introducing the bias to the gradient due to the presence of $\nabla log Z$. Normally calculating  $\nabla log Z$ requires sampling from the mode with every update of the parameters; it would be useful to provide exact update rule for clarity. Is the learning rule unbiased?

Since $\log c(x)^2 = 2 \log c(x)$ the majority of the difference between maximising MM and MM^2 comes from the difference in the gradients of $\log Z$ for MM and MM^2, I think this requires more clarity/explanation.

**Questions:**

See the weaknesses section.

---

> ### Author Response · Authors · 2023-11-17
> **Answer (1/2)**
>
> We thank the reviewer for appreciating our ideas and its potential. We believe our answers below can address all the raised concerns. Let us know if not, we are very keen to engage in a discussion.
>
> >[...] the presentation is very dense.
> >The authors should also elaborate on the selection of RG for improved clarity.
>
> We agree with the reviewer and improved the presentation. In the updated revision we have moved Section 4.2 to Appendix G, and better described how we build region graphs in the main text. We also provided more details in Appendix F and Figure F.1. Moreover, we splitted Section 3 in three parts to improve readability. If the reviewer has some additional reasonable suggestions, we will incorporate them in a new revision.
> >[...] for the cross comparison NPC^2(LT) vs MPC(BT) the latter can be better. This begs the question: is the RG doing the heavy lifting? If so, more empirical analysis would be helpful.
>
> Note that we already take into account the region graph (RG) construction as a latent confounder in Figure 4. And in all datasets, fixed a RG, NPC^2 is better than MPC. The case in which MPC (BT) is better than NPC^2 (LT) has been observed on Power only. If the reviewer can specify which particular aspect requires to be analysed empirically, we will gladly investigate it.
>
> >the result in Fig 5. on test data appears very small if statistically significant at all
>
> We agree with the reviewer, and believe our NPC^2s are overfitting in that experiment. Therefore, we updated the caption of Figure 5 and discussed this aspect in the text of the rebuttal revision. In general, we believe training NPC^2s can be more challenging (e.g., due to an observed higher sensitivity in parameters initialization), and we cannot rely on existing works investigating training as they assume non-negative parameters (such as [A]). See also our answer to Reviewer Bp5w.
>
> However, note that our experiments follow the same settings in [B], which is based on training a tractable probabilistic model (in our case a NPC^2) on sentences sampled from GPT2, with the aim of evaluating the expressiveness of our models on a more challenging scenario. So a higher train likelihood still brings enough evidence that NPC^2 are more expressive than MPCs. Furthermore note that, compared to [B], we use only half their number of sentences for temporal resource constraints (i.e. 4 millions instead of 8). In the limit and by sampling enough sentences from GPT2, one can definitely reduce the risk of overfitting. Lastly, we believe our work will also motivate future work on regularising NPC^2s as it is the first one to learn non-monotonic PCs on a large scale.
>
> [A] Liu et al., Scaling Up Probabilistic Circuits by Latent Variable Distillation, 2023
>
> [B] Zhang et al.,Tractable Control for Autoregressive Language Generation, 2023

---

> ### Author Response · Authors · 2023-11-17
> **Answer (2/2)**
>
> >a plot log-likelihood improvement vs CPU time would make the paper stronger
>
> In the latest revision we added Figure C.2 which exactly compares the actual computational cost of training the monotonic PCs and NPC^2 on UCI data sets and contrasts it with the gained likelihood. We show that NPC^2 adds little overhead during training in most configurations. In particular, Figure C.2 shows that the overhead for training on Gas results in NPC^2 achieving a x2 improvement on the final test log-likelihood. Moreover, on high-dimensional data such as BSDS300, NPC^2s are actually faster as they require fewer parameters and achieve higher log-likelihoods. We further analyze the “worst case” behavior on the largest circuits possible in Figure C.3 and confirm that the computation of Z for NPC^2 can match the cost of computing c(x) for MPC when amortized in batches.  See also our response to Reviewer 5G72.
>
> >Error bars in Figure 2 would help to understand the significance of empirical results.
>
> We assume you are referring to the table in Figure 4, let us know if not. We have already reported some of the standard deviations with multiple random repetitions in Table F.5 (which is now H.5 in the rebuttal revision). Apart from the results on Power, the standard deviations are quite small when compared to the average log-likelihoods and therefore support the significance of our empirical results.
>
> >the empirical improvements warranted by the proposed method are rather small and mostly shown on synthetic data.
>
> The experiments shown in Figure 4 are actually on real-world data sets. These data sets have been extensively used to evaluate deep generative models such as normalising flows in the past. To the best of our knowledge, our NPC^2 on these data sets score the current state-of-the-art for tractable probabilistic models supporting exact marginalization.
>
> Furthermore, the improvements on these data sets are actually significant. For example, we got a x2 improvement in terms of log-likelihoods on Gas (see table in Figure 4). Moreover, on MiniBooNE we went from -32.11 by monotonic PCs (MPC (BT)) to -26.92 by NPC^2 (BT), which is definitely much better if we also look at the multivariate Gaussian baseline that achieves -37.24.
>
> >Why the differences reported in F2(a) are so small?
>
> The log-likelihoods shown in F.2 (a) (in the rebuttal revision they are now shown in H.2) refer to mixture models having Categoricals as components. The small differences confirm our claim that negative parameters are not really useful if we use flexible yet expensive components, such as Categoricals (L309-320). We believe our empirical analysis is therefore balanced as it also provides failure cases for NPC^2s.
>
> >Can the authors elaborate on the statement 105-106 regarding batching?
>
> We want to remark that computing $Z$ and $\nabla \log Z$ can be done exactly in the proposed models (e.g., see Proposition 1), and therefore it does not require sampling. In our answer to Reviewer Lqff, we further discuss how the exact computation of $Z$ is performed in NPC^2s. Therefore, learning is performed by updating the parameters using exact gradients given by differentiating the negative log-likelihood on a batch of data. However, note that we need to differentiate $\log Z$ only once per batch, as it does not depend on the input, and therefore adds little overhead during training (see our response to Reviewer 5G72). Furthermore, note that gradients on MPCs and NPC^2s are different because NPC^2s allow for negative parameters.

---

> ### Author Response · Authors · 2023-11-20
>
> We hope we addressed all your concerns in our two comments above. If not, we are very keen to solve any additional issue that holds our submission back from a full acceptance.

---

### Author Response · Authors · 2023-11-22

We thank all reviewers for the time spent assessing the paper and recognizing the **originality and simplicity** of our method (“the paper proposes simple yet interesting idea” – nUGh, “it adds a new and provenly more expressive model” – Bp5w, “ it's a novel and very interesting [paper]” – uVYn, “I feel this is very intuitive and surprisingly simple” – Lqff), the importance of our **theoretical analysis** (“the related work develops important connections to other methods and models in the literature” – Bp5w, “extensive theoretical evaluation” – Lqff), and the overall significance of our **empirical evaluation** (“empirical results show better performance than baseline on some tasks” – 5G72, “often work better than mixture models and other alternatives such as flows” – uVYn).

We agree that the presentation could be improved (as pointed out by reviewers nUGh and Lqff), and more experimental details could be useful for readers (reviewers nUGh and 5G72). We believe the revised submission takes into account all the feedback from the reviewers. Below, we summarize the main modifications you can find in the latest revision:

- In Appendix C we added additional benchmarks evaluating the time and memory required to train NPC^2s. We show that NPC^2 adds little overhead during training in most configurations, as computing the partition function Z (i.e., the source of the overhead for NPC^2s) is comparable to evaluating the circuit on a batch of data. On some UCI data sets, training NPC^2s is even faster as they require less parameters to achieve the same (or better) log-likelihoods than monotonic PCs.
- Section 3 and Appendix F now discuss how region graphs are constructed in detail, and Figure F.1 illustrates examples of region graphs used in our experiments.
- We also split Section 3 into smaller parts focusing on different aspects, making the paper less dense.
- In Section 5, we added a sentence in the caption of Figure 5 mentioning the overfitting we observed with NPC^2s on the GPT2 distillation experiment. In the text, we added a few lines discussing the risk of overfitting in this setting and that possible ways to regularize NPC^2s can be explored in future works.
- To gain space for the above modifications, we moved part of our discussion in Section 4 on related works to Appendix G, as suggested by reviewer Lqff.
- We cited the works mentioned by reviewer Bp5w already using the signed log-sum-exp trick in their code bases.
- Our preliminary findings about the initialization of NPC^2s we discussed with reviewer Bp5w have been reported in Appendix H.3. Future directions investigating how to initialize, learn and regularize NPC^2s are interesting and deserve future work.

---

### Meta-Review · Area_Chair_MQpH · 2023-12-03

**Metareview:**

This paper explores the learning and inference of deep mixtures by squaring them. The paper demonstrates that squared circuits allow subtractions that can be exponentially more expressive than traditional additive mixtures. These claims are empirically validated on real-world distribution estimation tasks.

The reviewers think the paper makes some substantial novel contributions, both in terms and theory and also in terms of practical improvements over existing baselines. Most of the concerns raised by the reviewers are about the clarity of the presentation, which I believe can be addressed in a revision.  I therefore recommend acceptance.

**Justification For Why Not Higher Score:**

The topic is perhaps not mainstream enough for an oral presentation.

**Justification For Why Not Lower Score:**

I think this is a clear acceptance, but again the topic is perhaps not even mainstream enough for a spotlight.

---

### Decision · Program_Chairs · 2024-01-16

Accept (spotlight)